# Larger or Smaller Reward Margins to Select Preferences for LLM Alignment?

Kexin Huang [1]  Junkang Wu [1]  Ziqian Chen [2]  Xue Wang [2]  Jinyang Gao [2]  Bolin Ding [2]  Jiancan Wu [1]
Xiangnan He [1]  Xiang Wang [1]

## Abstract

Preference learning is critical for aligning large language models (LLMs) with human values, with the quality of preference datasets playing a crucial role in this process. While existing metrics primarily assess data quality based on either *explicit* or *implicit* reward margins, their single-margin focus often leads to contradictory evaluations for the same data. To address this issue, we propose a new metric of *alignment potential*, $M_{AP}$, which integrates both margins to quantify the gap from the model's *current implicit* reward margin to the *target explicit* reward margin, thereby estimating the model's potential to align on the preference data. Empirical results demonstrate that training on the data selected by $M_{AP}$ consistently enhances alignment performance, surpassing existing metrics across different base models and optimization objectives. Furthermore, our method can be extended to self-play data generation frameworks, where we use this metric to identify high-quality data within the self-generated content by LLMs. Under this data generation scenario, our method surpasses current state-of-the-art methods across various training settings and demonstrates continuous improvements with increasing dataset size and training iterations.

## 1. Introduction

Learning from human feedback is essential for aligning large language models (LLMs) (OpenAI, 2023; Touvron et al., 2023) with human preference, ensuring they are helpful, honest, and harmless (Askell et al., 2021). A standard method to achieve such alignment is reinforcement learning from human feedback (RLHF) (Christiano et al., 2017;

Ouyang et al., 2022; Stiennon et al., 2020), which involves iterative LLM fine-tuning and reward model training. To address the complexity inherent in this multi-stage training process, several offline methods — such as DPO (Rafailov et al., 2024) and SimPO (Meng et al., 2024) — have been proposed. They directly align LLMs using a static offline preference dataset $\{(x, y_w, y_l)\}$, where $y_w$ and $y_l$ denote the preferred (winning) and less preferred (losing) responses to the input prompt $x$, respectively. Nevertheless, due to the offline nature, these methods rely heavily on the quality of the preference dataset, which can substantially impact the alignment process (Liu et al., 2024a; Tajwar et al., 2024; Xiong et al., 2024; Yang et al., 2024).

Recent research has focused on improving LLM alignment by enhancing the quality of preference datasets, employing strategies such as selecting high-quality data for training (Khaki et al., 2024; Muldrew et al., 2024; Morimura et al., 2024) and re-weighting the loss based on data quality (Xu et al., 2023; Wang et al., 2024b; Wu et al., 2024). These approaches primarily rely on two metrics to assess the quality of preference pair $(x, y_w, y_l)$ in the offline dataset: the *explicit reward margin* metric (Wang et al., 2024b):

$$M_r(x, y_w, y_l) = |r(x, y_w) - r(x, y_l)|,$$

based on the explicit reward $r(x, \cdot)$ provided by an external reward model; and the *implicit reward margin* metric (Yang et al., 2024; Muldrew et al., 2024):

$$M_\pi(x, y_w, y_l) = |\hat{r}_\theta(x, y_w) - \hat{r}_\theta(x, y_l)|,$$

based on the implicit reward (Rafailov et al., 2024) $\hat{r}_\theta(x, \cdot) = \beta \log \frac{\pi_\theta(\cdot|x)}{\pi_{\text{ref}}(\cdot|x)}$ derived by the LLM policy $\pi_\theta(\cdot|x)$ and a reference policy $\pi_{\text{ref}}(\cdot|x)$. While it has been demonstrated that selecting data with *larger explicit reward margins* (Wang et al., 2024b; Ye et al., 2024) or *smaller implicit reward margins* (Yang et al., 2024) for training can improve alignment performance, these two metrics could provide conflicting guidance for the same data. To illustrate this conflict, we present two examples from the preference dataset used in SimPO (Figure 1a), where the same preference pair is deemed high quality by one metric but low quality by the other. Such inconsistency in data quality evaluations naturally raises a critical question:

[1]MoE Key Lab of BIPC, University of Science and Technology of China, Hefei, China [2]Independent Researcher. Correspondence to: Xiangnan He <xiangnanhe@gmail.com>, Xiang Wang <xiangwang1223@gmail.com>.

*Proceedings of the 42nd International Conference on Machine Learning*, Vancouver, Canada. PMLR 267, 2025. Copyright 2025 by the author(s).

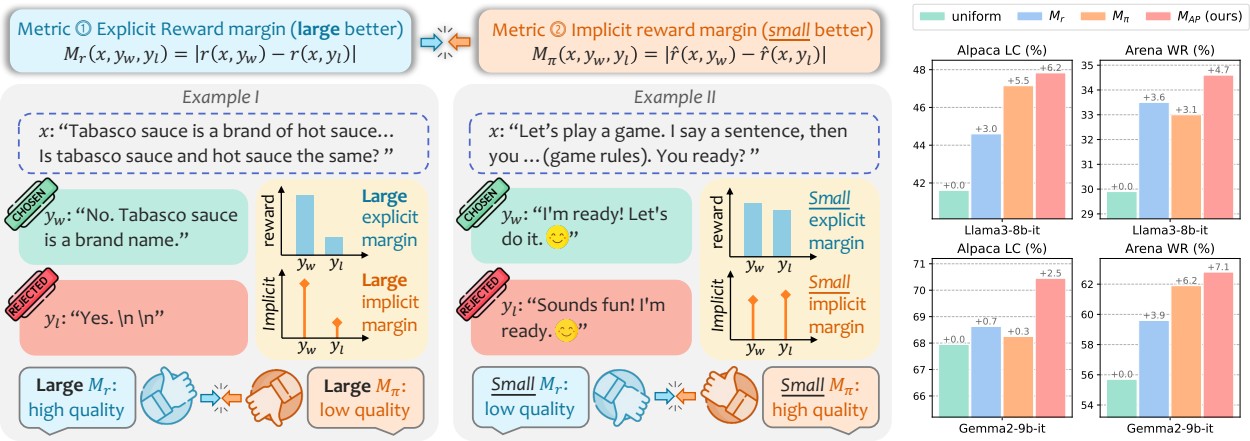

(a) Two examples illustrating the conflict between explicit and implicit reward margins      (b) Data selection performance

*Figure 1.* (1a) **Contradiction of existing metrics.** The left example, with large explicit and implicit reward margins, is rated as "high quality" by explicit reward margin but "low quality" by implicit reward margin. The right example, where both margins are small, is rated as "low quality" by the explicit margin but "high quality" by the implicit margin. In both cases, the implicit margins are already aligned with the explicit ones, making both data rated as "low quality" by our metric, *i.e.,* no need for further training. Existing metrics fail on these examples because they solely focus on a single margin, neglecting the crucial gap between them. (1b) **Enhanced performance by data selection.** Llama-3-8b-instruct and Gemma-2-9b-it's performance on Top-40% data subset selected by different data metrics (uniform refers to uniformly sampling 40% data from the dataset), with our proposed metric achieving the highest results.

*Is there a more reliable and theoretically grounded metric for evaluating preference data quality in alignment?*

In response to this question, we revisit the alignment objective in RLHF, where the optimum is characterized by Rafailov et al. (2024): $\hat{r}_\theta^*(x, y) = r(x, y) + c(x)$, with $c(x)$ being a partition term independent of $y$. This suggests that the implicit reward margin at optimality should be equivalent to explicit reward margin: $|\hat{r}_\theta^*(x, y_w) - \hat{r}_\theta^*(x, y_l)| \equiv |r(x, y_w) - r(x, y_l)|$. Building upon this insight, we propose the *alignment potential* metric, $M_{AP}$, which quantifies the gap from the current model's implicit reward margin to the target explicit reward margin:

$$M_{AP}(x, y_w, y_l)$$
$$= \underbrace{|r(x, y_w) - r(x, y_l)|}_{\text{target explicit reward margin}} - \underbrace{|\hat{r}_\theta(x, y_w) - \hat{r}_\theta(x, y_l)|}_{\text{current implicit reward margin}}. \quad (1)$$

This metric estimates the model's "potential" to align on the preference data by measuring how much the model can improve its preference discrimination: as shown in Equation (1), a preference pair $(x, y_w, y_l)$ with a *higher alignment potential* score indicates a significant reward improvement of the preferred response $y_w$ over $y_l$ while the model cannot sufficiently differentiate them (*i.e.,* similar implicit rewards), reflecting the large potential for alignment enhancement. To validate this, we conduct a preliminary experiment by benchmarking models trained on the top-rated data subsets evaluated by various metrics within SimPO's preference dataset. As depicted in Figure 1b, our proposed $M_{AP}$ metric outperforms other metrics, verifying its effectiveness in identifying high-quality data for alignment.

Furthermore, $M_{AP}$ not only supports selecting high-quality preference pairs in *existing* datasets, but also generalizes well to scenarios involving *additional* data, such as self-play alignment frameworks. In these frameworks, LLMs actively generate new training data through response sampling (Munos et al., 2024; Chen et al., 2024) or prompt creation (Ye et al., 2024). Our metric seamlessly integrates into such self-play frameworks, facilitating the identification of high-quality data within the intricate self-generated content. Consequently, it enables efficient expansion of high-quality preference data from a minimal seed dataset, thereby circumventing the constraints of static datasets and enhancing both alignment efficacy and efficiency.

We evaluate our proposed $M_{AP}$ metric through extensive experiments within such a self-play framework. Using $M_{AP}$, we first select top-rated subsets from the self-generated dataset for continued policy model training, achieving significant improvements over the current state-of-the-art results across various base models (*e.g.,* Llama (Dubey et al., 2024), Gemma (Team et al., 2024)) and learning algorithms (*e.g.,* DPO (Rafailov et al., 2024), SimPO (Meng et al., 2024)). Scaling up training by enlarging the generate-then-selected dataset further yields remarkable performance gains, even surpassing the model trained on twice the size of the UltraFeedback dataset. Additional experiments demonstrate that increasing training iterations under our metric's guidance leads to consistent performance gains over the models trained on the default dataset. These empirical results validate the practical utility of our proposed $M_{AP}$ metric across diverse training conditions, providing a robust solution for

enhancing LLM alignment through optimized preference data selection and utilization.

## 2. Preliminaries

**RLHF with reward models.** To align a supervised fine-tuned (SFT) model $\pi_{\text{SFT}}$ with human preferences, the standard RLHF pipeline begins with the reward modeling phase (Ziegler et al., 2019; Stiennon et al., 2020; Ouyang et al., 2022). A preference dataset $\mathcal{D} = \{(x, y_w, y_l)\}$ is constructed by sampling response pairs $(y_1, y_2) \sim \pi_{\text{SFT}}(\cdot|x)$ from prompts $x$, and human annotators evaluate the preference $y_w \succ y_l|x$, where $y_w$ and $y_l$ are the preferred (winning) and less preferred (losing) responses, respectively. Although the latent reward $r^*(x, y)$ underlying human preference is unknown, the Bradley-Terry (BT) model (Bradley & Terry, 1952) provides an effective proxy to capture human preference probabilities $p^*$ as follows:

$$p^*(y_1 \succ y_2|x) = \sigma(r^*(x, y_1) - r^*(x, y_2)), \quad (2)$$

where $\sigma(\cdot)$ denotes the logistic function. Therefore, a reward model (RM) $r_\phi(x, y)$ can be parameterized and trained with the preference dataset $\mathcal{D}$ via a negative log-likelihood loss:

$$\mathcal{L}_R(\phi) = -\mathbb{E}_{(x,y_w,y_l)\sim\mathcal{D}} \left[\log \sigma(r_\phi(x, y_w) - r_\phi(x, y_l))\right]. \quad (3)$$

With the learned reward model providing feedback, the language model $\pi_\theta$ is subsequently optimized via RL using the KL-regularized objective:

$$\max_{\pi_\theta} \mathbb{E}_{\substack{x\sim\mathcal{D} \\ y\sim\pi_\theta(\cdot|x)}} [r_\phi(x, y)] - \beta\mathbb{D}_{\text{KL}}(\pi_\theta(y|x)\|\pi_{\text{ref}}(y|x)), \quad (4)$$

where $\pi_{\text{ref}}$, typically instantiated via the SFT model $\pi_{\text{SFT}}$, serves as the reference policy and $\beta$ is a hyperparameter controlling the deviation from the reference model.

**Direct Preference Optimization (DPO).** DPO (Rafailov et al., 2024) is a widely adopted offline preference optimization method. Instead of learning an external reward model, DPO reparameterizes the reward function $r(x, y)$ using a closed-form solution to the KL-regularized reward maximization problem:

$$r(x, y) = \beta \log \frac{\pi_r(y|x)}{\pi_{\text{ref}}(y|x)} + \log Z(x), \quad (5)$$

where $Z(\cdot)$ serves as the partition function independent of $y$. Applying this reparameterization to the current LLM $\pi_\theta$, DPO derives an implicit reward $\hat{r}_\theta(x, y) = \beta \log \frac{\pi_\theta(y|x)}{\pi_{\text{ref}}(y|x)}$. By fitting it with the offline preference dataset like Equation (3), the DPO loss is defined as:

$$\mathcal{L}_{\text{DPO}}(\theta) = -\mathbb{E}_{(x,y_w,y_l)\sim\mathcal{D}} \left[\log \sigma(\hat{r}_\theta(x, y_w) - \hat{r}_\theta(x, y_l))\right]. \quad (6)$$

**Simple Preference Optimization (SimPO).** SimPO (Meng et al., 2024) builds on DPO by introducing two key modifications to achieve SOTA performance. First, it proposes a new implicit reward $\hat{r}_\theta^{\text{Sim}}(x, y) = \beta \log \pi_\theta(y|x)/|y|$ to eliminate the need for a reference model, where $|y|$ denotes the length of response $y$. Such a formulation can be seen as a length-normalized special case of DPO's implicit reward with a uniform reference model (Wu et al., 2025a). Second, it introduces a classification margin term $\gamma$ into the objective. Consequently, the SimPO loss is formulated as follows:

$$\mathcal{L}_{\text{SimPO}}(\theta) = \quad (7)$$
$$- \mathbb{E}_{(x,y_w,y_l)\sim\mathcal{D}} \left[\log \sigma(\hat{r}_\theta^{\text{Sim}}(x, y_w) - \hat{r}_\theta^{\text{Sim}}(x, y_l) - \gamma)\right].$$

**Data quality metrics.** Here we study the data quality evaluation problem in offline preference optimization. We denote the metric by $M(x, y_w, y_l; \theta)$ to assess the quality of preference data $(x, y_w, y_l)$ for the LLM model $\pi_\theta$. To evaluate different metrics, we can benchmark models trained with the top-$k$ subset of the original preference dataset (Yang et al., 2024), denoted as $\mathcal{D}_k := \{(x, y_w, y_l)|M(x, y_w, y_l; \theta) \geq \tau_k\}$, where $\tau_k$ is the $k$-th highest score within the dataset. For model training, we mainly employ DPO and SimPO as optimization methods due to their verified efficacy.

## 3. Alignment Potential Metric

This section introduces our alignment potential metric, devised to assess the quality of preference data $(x, y_w, y_l)$ for alignment. §3.1 elaborates on the formulation of alignment potential metric. §3.2 empirically evaluates the performance of various metrics by using them to select preference data for training. §3.3 provides a theoretical analysis justifying the proposed metric's ability to identify high-quality data for model alignment.

### 3.1. Metric Formulation

Preference optimization methods (*e.g.,* DPO, SimPO) leverage preference datasets to transform unaligned LMs into models that align with human values. A natural principle for evaluating the quality of preference data $(x, y_w, y_l)$ is to measure how much the model can improve its alignment with human preferences by learning from such data. This can be essentially quantified by measuring the discrepancy between the current model and an aligned optimal state on that data. For the aligned optimal policy $\pi_\theta^*$, Equation (5) describes an optimal condition:

$$r^*(x, y) = \hat{r}_\theta^*(x, y) + \log Z(x),$$

which leads to the following equivalence for any preference data $(x, y_w, y_l)$:

$$r^*(x, y_w) - r^*(x, y_l) \equiv \hat{r}_\theta^*(x, y_w) - \hat{r}_\theta^*(x, y_l). \quad (8)$$

Based on this, the discrepancy between the current model and the aligned optimal on any given preference data can be quantified by the $\ell_1$-distance, resulting in the $M_1$ metric:

$$M_1(x, y_w, y_l; \theta, r^*) \tag{9}$$
$$:= |(\hat{r}_\theta^*(x, y_w) - \hat{r}_\theta^*(x, y_l)) - (\hat{r}_\theta(x, y_w) - \hat{r}_\theta(x, y_l))|$$
$$= |\underbrace{(r^*(x, y_w) - r^*(x, y_l))}_{\text{given by aligned optimum } \pi_\theta^*} - \underbrace{(\hat{r}_\theta(x, y_w) - \hat{r}_\theta(x, y_l))}_{\text{given by current model } \pi_\theta}|.$$

Here the term $\hat{r}_\theta(x, y_w) - \hat{r}_\theta(x, y_l)$ represents the reward margin (without absolute value) between $y_w$ and $y_l$ as evaluated by the model $\pi_\theta$, indicating the model's preference judgment between $y_w$ and $y_l$ (Rafailov et al., 2024; Muldrew et al., 2024). Consequently, $M_1$ measures the gap between preferences given by the current model $\pi_\theta$ and the aligned optimum $\pi_\theta^*$ for the data $(x, y_w, y_l)$. With this metric, a larger gap indicates greater potential for improvement (*i.e.,* high-quality data), while a smaller gap suggests less need for enhancement (*i.e.,* low-quality data).

Nevertheless, direct computing $M_1$ requires access to the latent, inaccessible human preference reward $r^*$. To make this metric practically applicable, we propose using a reward model $r(\cdot)$ for guidance and introduce two key adaptations for practical offline preference learning: *unidirectional calibration* and *reward noise regularization*.

**Unidirectional calibration.** In typical offline preference learning scenarios (*e.g.,* DPO), preferences $y_w \succ y_l | x$ in the dataset are annotated by an external reward model such that $r(x, y_w) > r(x, y_l)$, rather than being sampled from real human preference probabilities as in Equation (2). Given this deterministic nature, objectives like DPO and SimPO adopt a *unidirectional* update strategy to consistently increase the margin $\hat{r}_\theta(x, y_w) - \hat{r}_\theta(x, y_l)$ (*cf.* Equation (6)), as opposed to regressing the target aligned optimum $\hat{r}^*(x, y_w) - \hat{r}^*(x, y_l)$ (Rosset et al., 2024). In contrast, the absolute value in $M_1(x, y_w, y_l)$ function measures a *bidirectional* gap relative to the centering optimum $r^*(x, y_w) - r^*(x, y_l)$. To reconcile this, we apply a unidirectional calibration by omitting the absolute value from $M_1(x, y_w, y_l)$:

$$M_+(x, y_w, y_l; \theta, r^*) \tag{10}$$
$$:= (r^*(x, y_w) - r^*(x, y_l)) - (\hat{r}_\theta(x, y_w) - \hat{r}_\theta(x, y_l)).$$

Large $M_+(x, y_w, y_l)$ values thus highlight preference data where $\hat{r}_\theta(x, y_w) - \hat{r}_\theta(x, y_l)$ is small, implying a need for increases consistent with offline preference methods.

**Reward noise regularization.** Moreover, substituting $r^*$ with $r$ can introduce reward model noise (Casper et al., 2023), which might lead to erroneous preference annotations (Stiennon et al., 2020; Gao et al., 2023). To illustrate this phenomenon, we present a preference data example from SimPO's dataset in Figure 2 (further details are provided

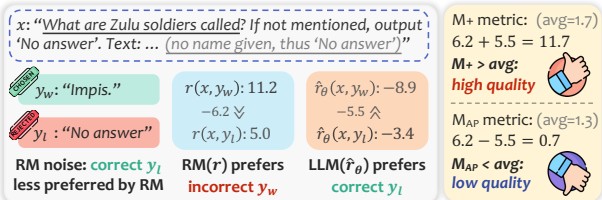

*Figure 2.* Reward noise example from SimPO's dataset, where the less preferred response $y_l$ by RM is the correct answer. $M_+$ will mislabel this data as high-quality, which the $M_{AP}$ metric avoids.

in Table 12 in Appendix E). In this example, the reward model erroneously assigns a higher reward to the incorrect answer. In contrast, the implicit reward given by the LLM $\pi_\theta$ in this example correctly identifies the preferable answer, reflecting LLM's ability to discern certain preferences, as supported by recent studies (Chen et al., 2025; Xu et al., 2024b; Kim et al., 2025).

This inspires us to revisit the $M_+$ metric. Clearly, it tends to prioritize data where $r(x, y_w) - r(x, y_l) \gg 0$ (*i.e.,* $y_w$ is strongly preferred by RM) while $\hat{r}_\theta(x, y_w) - \hat{r}_\theta(x, y_l) \ll 0$ (*i.e.,* $y_w$ is strongly disliked by LLM). While such strong contradictions may suggest mistaken preference judgments by the LLM that necessitate alignment training, they also risk amplifying faulty preference annotations caused by the reward model noise. Considering the aforementioned example again where the reward model assigns incorrect rewards, the RM assigns $r(x, y_w) - r(x, y_l) = 6.2$ while the LLM evaluates $\hat{r}_\theta(x, y_w) - \hat{r}_\theta(x, y_l) = -5.5$, resulting in an inflated $M_+$ score of 11.7, nearly seven times the dataset's average $M_+$ score. As a result, this noisy data would be incorrectly rated as "high quality" by $M_+$.

The root issue lies in the contradictions between reward model annotations and LLM judgments, which may signal reward model noise when the LLM can accurately discern certain preferences. To address this, we revise M+ by reintroducing absolute values as a regularizer, leading to the proposed alignment potential metric:

$$M_{AP}(x, y_w, y_l; \theta, r)$$
$$:= \underbrace{|r(x, y_w) - r(x, y_l)|}_{\substack{\text{target explicit reward margin} \\ \text{(by aligned optimum } \pi_\theta^*)}} - \underbrace{|\hat{r}_\theta(x, y_w) - \hat{r}_\theta(x, y_l)|}_{\substack{\text{current implicit reward margin} \\ \text{(by current model } \pi_\theta)}} \tag{11}$$

This adjustment ensures that data with high $M_{AP}$ scores indicate substantial reward improvements, while the LLM shows uncertainty in preference determination (*i.e.,* similar implicit rewards). It better identifies data with alignment potential and avoids unreliable contradictory annotations.

For the previous example, our alignment potential metric yields a significantly lower score $M_{AP} = 0.7$, below the dataset's average $\overline{M}_{AP} \approx 1.3$. Consequently, this noisy preference data is rated as "low quality" by the $M_{AP}$ met-

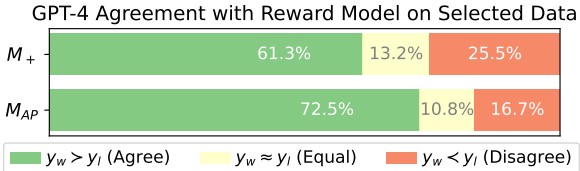

*Figure 3.* GPT-4 agreement with reward model's annotation on data selected by $M_+$ or $M_{AP}$ metric from SimPO's dataset. The data selected by $M_{AP}$ has a notably higher agree rate than $M_+$, indicating less reward annotation noise.

ric, offering robust data quality evaluation against reward noise. Furthermore, we validate the regularization term in $M_{AP}$ by applying both $M_+$ and $M_{AP}$ metrics to rate and select top-10% preference pairs in SimPO's dataset and prompting GPT-4 to reassess the selected pairs against the original reward-model annotations. As shown in Figure 3, pairs selected by $M_{AP}$ exhibit significantly higher agreement with GPT-4, indicating reduced annotation noise[1], and evidencing the regularization's effectiveness in $M_{AP}$.

**Relationship with existing metrics.** As shown in Equation (11), the proposed alignment potential metric can be split into two components: the target *explicit reward margin* term $M_r = |r(x, y_w) - r(x, y_l)|$, derived from aligned optimum $\pi_\theta^*$, and the current *implicit reward margin* term $M_\pi = |\hat{r}_\theta(x, y_w) - \hat{r}_\theta(x, y_l)|$, given by current model $\pi_\theta$. While the $M_{AP}$ metric measures the gap from the current implicit reward margin to the target explicit reward margin, thereby serving as an indicator for alignment potential and data quality, each of these margins has been independently utilized as a data quality metric. Empirical studies have validated that selecting data with *larger* $M_r$ (Wang et al., 2024b; Ye et al., 2024) or *smaller* $M_\pi$ (Yang et al., 2024) for training improves alignment performance, which is also consistent with the trend described in $M_{AP}$. Additionally, as $\hat{r}_\theta(x, y) = \beta \log \frac{\pi_\theta(y|x)}{\pi_{\text{ref}}(y|x)}$ contains a hyper-parameter $\beta$, the ranking induced by $M_{AP}$ converges with $M_r$ as $\beta \to 0$, and with $-M_\pi$ as $\beta \to \infty$. Therefore, the $M_{AP}$ metric offers an integrated metric unifying both the explicit and implicit reward perspectives. More importantly, it underscores the potential for alignment by assessing the discrepancy between target and current reward margins, rather than relying on a singular metric.

## 3.2. Empirical Evaluation for Data Metrics

In this section, we conduct a preliminary experiment to evaluate different data quality metrics, using them to select preference data for alignment training. The results demonstrate the superior ability of our metric to select high-quality data, leading to improved alignment performance across

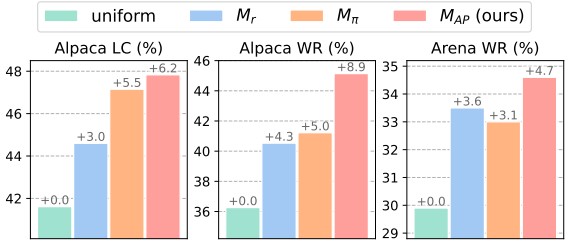

*Figure 4.* Performance of Llama-3-8b-instruct model trained on preference pairs selected by different metrics.

multiple benchmarks and models.

**Models and datasets.** We focus on preference learning for the Llama-3-8b (Dubey et al., 2024) and Gemma-2-9b (Team et al., 2024) models. Following Meng et al. (2024), we employ pre-trained instruction-tuned models as SFT models and utilize the same preference datasets in SimPO: llama3-ultrafeedback-armorm and gemma2-ultrafeedback-armorm, for llama and gemma models respectively. These datasets are generated by sampling responses from LLMs for prompts in the UltraFeedback dataset (Cui et al., 2024) and then annotating preferences with the ArmoRM reward model (Wang et al., 2024a). Additionally, SimPO provides another llama-based dataset with the preferences annotated using the PairRM reward model (Jiang et al., 2023): llama3-ultrafeedback, which we include as an ablation study on different reward models.

**Data quality metrics.** Various data quality metrics are utilized to score preference pairs and select the top-40% subset (approximately 24k pairs) from the SimPO datasets for further training. We compare the proposed alignment potential metric $M_{AP}$ against the explicit reward margin $M_r$ metric, the negative[2] implicit reward margin $-M_\pi$ metric, and a uniform baseline that randomly samples 40% of the data for training. Additionally, when calculating data metrics before training, the model $\pi_\theta$ used for calculating implicit rewards $\hat{r}_\theta$ is initialized from the reference model $\pi_{\text{ref}}$, making the DPO's implicit reward $\hat{r}_\theta = \beta \log \frac{\pi_\theta(y|x)}{\pi_{\text{ref}}(y|x)}$ constant zero. To address this, we utilize the SimPO's implicit reward $\hat{r}_\theta^{\text{Sim}}(x, y) = \beta \log \pi_\theta(y|x)/|y|$ for practical implementation across this paper. Further implementation details are provided in Appendix D.

**Training and Evaluation.** We primarily adopt SimPO for preference optimization and include DPO and IPO (Azar et al., 2024) as an ablation study concerning different optimization objectives. After preference learning, evaluations are conducted using two widely recognized open-ended instruction-following benchmarks: AlpacaEval 2 (Dubois et al., 2024) and Arena-Hard (Li et al., 2024). These benchmarks assess the model's general conversational abilities

---

[1]While GPT-4 might also make mistakes, alignment with its preferences suggests at least a lower likelihood of annotation noise.

[2]Since $M_\pi$ prioritizes data with smaller scores, we add a negative sign for consistency with the top-$k$ selection procedure.

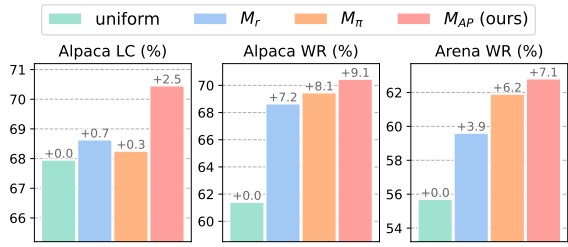

*Figure 5.* Performance of Gemma-2-9b-it model trained on preference pairs selected by different metrics.

across diverse queries and are extensively used in the research community. Our results report the win rate for Arena-Hard and both length-controlled (LC) and raw win rates (WR) for AlpacaEval 2.

**Empirical Results.** Our alignment potential metric consistently outperforms existing data quality metrics: As depicted in Figures 4 and 5, while all three metrics can enhance the alignment performance, our alignment potential metric $M_{AP}$ achieves the highest performance across all benchmarks and models. Particularly, models trained on data selected via our metric significantly outperform the uniform baseline by 2.5 and 6.2 points on Alpaca LC, and by 4.7 and 7.1 points on Arena WR. Such notable performance enhancement underscores the effectiveness and robustness of our metric in identifying high-quality preference data for alignment training.

**Ablation Study.** We conduct in-depth ablation studies to rigorously evaluate various data quality metrics under different reward models, training objectives, and amount of selected data (*cf.* Appendix B.1). As illustrated in those experiment results, our alignment potential metric consistently surpasses existing baselines even with these variations, further validating its robustness and efficacy.

### 3.3. Theoretical Analysis

Building upon the empirical results showcasing enhanced alignment with data selected by our alignment potential metric, we now provide a theoretical explanation for its effectiveness. With the preference learning process transforming unaligned LLM $\pi_\theta$ into the aligned optimum $\pi_\theta^*$, our proposed metric, which selects preference data with a larger gap between $\pi_\theta$ and $\pi_\theta^*$, serves as an adversarial sampler prioritizing "hard" examples for training. Compared with uniform sampling from the dataset, such a philosophy can intuitively speed up preference learning by prioritizing data with the largest optimization gaps, which requires the most update, to train on. Below, we rigorously substantiate this intuition through the lens of stochastic optimization.

**Contextual bandit setting.** Following the standard theoretical frameworks in RLHF (Rafailov et al., 2024; Munos

et al., 2024; Rosset et al., 2024), we reformulate the problem as a contextual bandit, with a context (prompt) space $\mathcal{X}$, an arm (response) space $\mathcal{Y}$ and a reward function $r : \mathcal{X} \times \mathcal{Y} \to [0, 1]$. A policy (LLM) $\pi : \mathcal{X} \to \Delta\mathcal{Y}$ maps each context to a probability simplex over the arm space, which is parameterized with $\theta : \mathcal{X} \times \mathcal{Y} \to \mathbb{R}$ using the softmax transformation (Mei et al., 2020) akin to LLMs: $\pi_\theta(y|x) = \frac{\exp(\theta(x,y))}{\sum_{y' \in \mathcal{Y}} \exp(\theta(x,y'))}$. The optimization objective is to maximize the KL-regularized reward, as expressed in Equation (4), with the optimal condition given by: $r(x, y) = \hat{r}_{\theta^*}(x, y) + \log Z(x)$. To quantify the distance between $\theta$ and $\theta^*$ for theoretical analysis, we utilize the $M_1(x, y, y'|\theta, r)$ metric, which measures the discrepancy between $\pi_\theta$ and optimal policy $\pi_{\theta^*}$ on any preference pairs $(x, y, y')$, and define the following error function:

$$\text{Dist}(\theta, \theta^*) = \sqrt{\frac{1}{|\mathcal{X}||\mathcal{Y}|^2} \sum_{x \in \mathcal{Y}} \sum_{y, y' \in \mathcal{Y}} M_1(x, y, y'; \theta, r)^2}.$$

**Optimization methods.** We investigate convergence rates of different sampling strategies when optimizing the policy $\pi_\theta$ using the DPO loss, whose population-form is defined as (Azar et al., 2024):

$$\min_{\pi_\theta} \mathbb{E}_{(x,y,y') \sim s}[-p(y \succ y'|x) \log \sigma(\hat{r}_\theta(x, y) - \hat{r}_\theta(x, y'))],$$

where $s : \mathcal{X} \times \mathcal{Y}^2 \to [0, 1]$ represents the sampling probability, and $p(y \succ y'|x) = \sigma(r(x, y) - r(x, y'))$ denotes the preference given by BT model. We apply stochastic gradient descent to optimize the DPO loss and compare the performance of two samplers: a uniform sampler $s_u(x, y, y') = \frac{1}{|\mathcal{X}||\mathcal{Y}|^2}$ and an adversarial sampler that selects preference data $(x, y, y')$ with the highest $M_1(x, y, y')$ score. Note that we directly employ the original $M_1$ metric within this theoretical setting, with no need for other calibrations specially designed for practical offline preference learning scenarios (*i.e.,* $M_+$ or $M_{AP}$).

**Convergence rates.** Our theoretical analysis reveals that by selecting preference data $(x, y, y')$ with the largest gap, *i.e.,* the $M_1(x, y, y')$ score, the adversarial sampler achieves more than 2 times faster convergence than the uniform sampler. Formally, we state the following theorem (the proof is provided in Appendix C.1):

**Theorem 3.1.** *Let $T_u(\varepsilon)$ and $T_{adv}(\varepsilon)$ be the (expected) iterations required for the error $\text{Dist}(\theta^t, \theta^*)$ to reduce to $\varepsilon\text{Dist}(\theta^0, \theta^*)$ under uniform and adversarial sampling, respectively. With optimal learning rates, we have:*

$$T_{adv}(\varepsilon) < 0.5 T_u(\varepsilon).$$

We also conduct numerical experiments to validate this theoretical result. As illustrated in Figure 10, the error reduction rate with the adversarial sampler significantly outperforms

that of the uniform sampler, thereby confirming the theoretical acceleration. More importantly, such an acceleration provides a theoretical justification for selecting data with higher alignment potential scores for training.

## 4. High-Quality Data Generation

The proposed data metric extends beyond merely selecting high-quality data subsets in *existing* datasets; it also applies to *additional* data scenarios, *i.e.,* to generate more high-quality data for alignment training. To facilitate this, we integrate our proposed alignment potential metric within the evolving alignment (eva) framework (Ye et al., 2024), a self-play data generation technique that employs LLM to generate new prompts $x$ and associated response pairs $(y_w, y_l)$ to augment high-quality preference data.

Similar to SimPO, the eva framework begins by constructing a preference dataset $\mathcal{D} = \{(x, y_w, y_l)\}$ by sampling LLM's responses $(y_1, y_2)$ to an existing prompts set $x \in \mathcal{X}$ (*e.g.,* UltraFeedback (Cui et al., 2024)), and annotating preferences with a reward model such that $r(x, y_w) > r(x, y_l)$. Then eva augments a high-quality subset within this dataset through a **select-then-evolve** pipeline, which can be delineated as follows:

1. Initially, eva evaluates each preference pair $(x, y_w, y_l)$ in $\mathcal{D}$ using the explicit reward margin metric $M_r$. The top-$k$ subset $\mathcal{D}_k$ is then selected based on their $M_r$ scores.

2. Subsequently, eva leverages the LLM itself to rewrite selected prompts $\mathcal{X}_k$ in $\mathcal{D}_k$, yielding a set of newly generated prompts $\mathcal{X}'_k$. Therefore a new preference dataset $\mathcal{D}' = \{(x', y'_w, y'_l)\}$ can be generated with a similar sample and annotation process as $\mathcal{D}$.

To generate new prompts $\mathcal{X}'_k$, eva employs the off-the-shell instructions from EvolInstruct (Xu et al., 2024a), a popular method for autonomous prompt synthesis with LLM, which involves querying the LLM for in-depth and in-breadth evolving of existing prompts (*cf.* Appendix D). The data generation methodology above allows eva to augment data selected by $M_r$ and generate additional preference datasets for alignment training.

Our proposed $M_{AP}$ metric can be seamlessly integrated into this framework for identifying high-quality data, by replacing the $M_r$ metric during the prompts selection phase of eva. Nonetheless, the subsequent data evolving process does not currently involve data quality evaluation through any metric, thus failing to guarantee the quality of the evolved dataset. To address this shortcoming, we modify this self-play procedure by reordering the two phases of eva: We first evolve the existing prompts $\mathcal{X}$ in $\mathcal{D}$ to derive additional dataset $\mathcal{D}'$ and then select a high-quality subset $\mathcal{D}'_k$ from the newly evolved dataset. This **evolve-then-select** modification ensures that the resultant dataset $\mathcal{D}'_k$ aligns with our data quality metric,

offering a more rigorous validation of our metric in the data generation scenarios. Additionally, this data generation and selection process can be iteratively conducted akin to eva, which we formally detail in Algorithm 1.

---

**Algorithm 1** Eva with *Evolve-then-Select* Pipeline

---

**Input:** SFT model $\pi_{\text{SFT}}$, reward model $r$, existing prompts $\mathcal{X}$ and data quality metric $M$.
Initialize $\pi_{\theta_0}$ with $\pi_{\text{SFT}}$.
**for** iteration $t = 1, \ldots, T$ **do**
    preference dataset: $\mathcal{D} \leftarrow \text{GenDataset}(\mathcal{X}, \pi_{\theta_{t-1}}, r)$.
    evolved prompts: $\mathcal{X}' \leftarrow \text{EvolInstruct}(\mathcal{X})$.
    additional dataset: $\mathcal{D}' \leftarrow \text{GenDataset}(\mathcal{X}', \pi_{\theta_{t-1}}, r)$.
    top-$k$ dataset selected by $M$ metric:
        $\mathcal{D}'_k = \{(x, y_w, y_l) \in \mathcal{D} | M(x, y_w, y_l; \theta_{t-1}) \geq \tau_k\}$,
        with $\tau_k$ being the $k$-th largest $M$ score in $\mathcal{D}'$.
    preference optimization on $\mathcal{D} \cup \mathcal{D}'_k$:
        $\theta_t \leftarrow \theta_{t-1} - \eta \nabla_\theta \mathcal{L}(\theta_{t-1}; \mathcal{D} \cup \mathcal{D}'_k)$.
**end for**
**Return:** optimized policy $\pi_{\theta_T}$

---

## 5. Experiments

In this section, we present the experimental results utilizing our alignment potential metric within the context of the eva self-play data generation framework. Following eva, we use the UltraFeedback dataset (Cui et al., 2024) as our source of initial prompts and adopt the ArmoRM (Wang et al., 2024a) reward model for preference annotation. Similar to the setup in §3.2, we employ SFT models from the Llama-3-8b and Gemma-2-9b model families, apply SimPO and DPO loss for alignment training, and utilize AlpacaEval 2 and Arena Hard benchmarks for model evaluations.

### 5.1. Main Results

We first compare models trained with different datasets under the single-iteration setting. Utilizing the procedure outlined in Algorithm 1, we begin with 10k prompts from the UltraFeedback dataset to generate the initial uf-10k dataset $\mathcal{D}$. Then we evolve new prompts $\mathcal{X}'$ and construct the additional preference dataset $\mathcal{D}'$. Based on this new dataset, we select the top-10k subset using eva and our alignment potential metric, resulting in the $\mathcal{D}'_{\text{eva-10k}}$ and $\mathcal{D}'_{\text{ours-10k}}$ datasets. To evaluate the effectiveness of evolve-selected data against existing datasets, we create an additional dataset, $\mathcal{D}_{\text{uf-10k}}$, using supplementary 10k prompts from UltraFeedback.

Table 1 illustrates the alignment performance across models trained on various datasets. Here, "uf-10k" refers to the initial dataset $\mathcal{D}$, while "+uf-10k", "+eva-10k" and "+ours-10k" denotes models trained on the union of $\mathcal{D}$ with $\mathcal{D}_{\text{uf-10k}}$, $\mathcal{D}'_{\text{eva-10k}}$, and $\mathcal{D}'_{\text{ours-10k}}$ respectively. As shown in the table, while all three additional datasets contribute to improved

*Table 1.* Main results under the *evolve-then-select* self-play setting. Selecting data with the alignment potential metric for training, our method achieves notable alignment improvements, outperforming the UltraFeedback data (uf) and current SOTA method (eva).

| Method | Llama-3-8b-instruct | | | Gemma-2-9b-it | | |
|---|---|---|---|---|---|---|
| | AH | AE 2.0 | | AH | AE 2.0 | |
| | WR | LC | WR | WR | LC | WR |
| SFT | 21.4 | 23.25 | 23.50 | 40.7 | 48.75 | 36.49 |
| DPO$_{uf-10k}$ | 23.6 | 29.80 | 28.32 | 46.0 | 54.00 | 46.11 |
| +uf-10k | 35.0 | 42.67 | 41.67 | 56.4 | 62.56 | 62.17 |
| +eva-10k | 31.4 | 39.97 | 40.74 | 56.0 | 63.32 | 62.10 |
| **+ours-10k** | **35.4** | **42.82** | **45.00** | **56.6** | **63.88** | **64.52** |
| SimPO$_{uf-10k}$ | 24.2 | 28.45 | 26.23 | 45.2 | 56.90 | 43.80 |
| +uf-10k | 28.5 | 35.75 | 32.82 | 53.5 | 66.43 | 64.55 |
| +eva-10k | 32.6 | 41.12 | 39.94 | **61.0** | 66.78 | 66.86 |
| **+ours-10k** | **34.6** | **43.76** | **42.00** | 60.9 | **66.85** | **68.34** |

alignment performance, datasets selected via our metric consistently achieve superior results across different base models and training objectives, surpassing both existing datasets and the current SOTA method, eva. These consistent improvements underscore the effectiveness and robustness of our alignment potential metric in discerning high-quality data for alignment training.

Furthermore, we conduct additional experiments employing eva's original *select-then-evolve* pipeline, which initially selects a high-quality subset $\mathcal{D}_k$ for data evolving into the additional dataset $\mathcal{D}'_k$ (more details can be found in Appendix B.3). While the resultant $\mathcal{D}'_k$ dataset's quality may not match that of $\mathcal{D}_k$ due to the intricate data-evolving process, our experiments, as shown in Table 7, reveal that evolved datasets based on our metric's selected subsets still achieve the best overall performance. This suggests that high-quality initial subsets might positively influence the quality of the evolved data, thus enhancing the model alignment. Specifically, we illustrate this correlation between model performance and the alignment potential score of their corresponding datasets in Figure 9 (*cf.* Appendix B.3). The selected subset $\mathcal{D}_k$ using our metric yields a higher $M_{AP}$ score in the resultant $\mathcal{D}'_k$, and the model performance consistently improves with this score, affirming enhanced alignment through high-quality data generation.

### 5.2. Scaling with Dataset Size

Since the main experiments involve only 20k data, we extend our investigation to larger datasets to assess the scalability of the proposed metric. Within the *evolve-then-select* framework, we leverage LLMs to generate additional preference data based on 20k seed prompts from UltraFeedback and select a top-$k$ subset from the combined dataset with our alignment potential metric for SimPO training. For comparison, a full-size dataset built upon the entire UltraFeedback

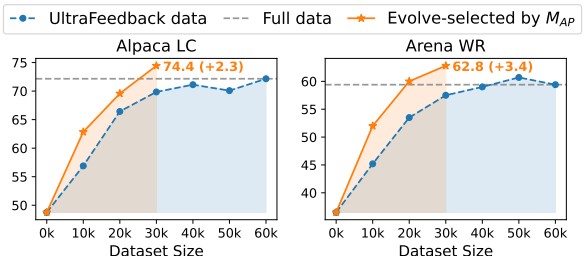

*Figure 6.* Performance of Gemma models as the dataset size increases.

prompts, comprising approximately 60k data, is randomly sampled with varied sizes for training.

Figure 6 illustrates the model performance relative to dataset size. In contrast to the gradual performance increase observed with the full UltraFeedback dataset, datasets selected using our metric demonstrate significantly faster improvements. Remarkably, training on a 30k subset containing only half the data leads to superior performance compared to the full UltraFeedback dataset, improving 2.3 points in Alpaca LC and 3.4 points in Arena Hard. These substantial gains highlight the efficacy of our metric for selecting high-quality data, suggesting a potent avenue for LLM alignment with superior models even using smaller datasets.

### 5.3. Scaling with Training Iteration

Our metric is also applied to multi-iteration settings. Like the main experiment, we utilize a fixed 10k subset $\mathcal{X}$ from UltraFeedback to generate the initial dataset $\mathcal{D}$ for each iteration, and augment it with 10k additional evolve-selected data $\mathcal{D}'_{ours-10k}$, or supplementary data from UltraFeedback $\mathcal{D}_{uf-10k}$, for SimPO training. We report results for generating $\mathcal{D}_{uf-10k}$ by both fixed and additional prompts from UltraFeedback. Figure 7 illustrates the continuously increasing performance with more iterations when using our metric, which consistently outperforms the UltraFeedback data, even when extra data are sampled.

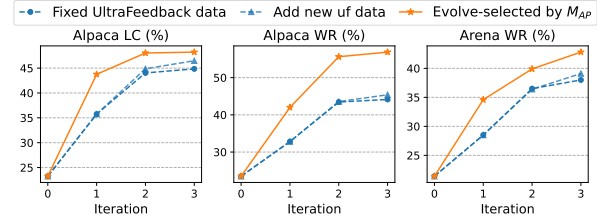

*Figure 7.* Performance of Llama models trained under iterative preference learning setting.

## 6. Conclusion and Future Work

In this work, we propose the *alignment potential* metric, $M_{AP}$, to evaluate preference data quality in alignment. By

measuring the gap from the model's current implicit reward margin to the target explicit reward margin, $M_{AP}$ quantifies the discrepancy between the current model and the aligned optimum, thereby indicating the potential for alignment enhancement. Extensive experiments validate the efficacy of $M_{AP}$ across various training settings under offline and self-play preference learning scenarios.

**Limitations and future work**. Despite the performance improvements, $M_{AP}$ requires tuning a parameter $\beta$ to combine the explicit and implicit margins; future work could explore how to set this ratio automatically. As for the models, we mainly investigate the alignment performance of instruction-tuned models, *i.e.,* the Llama-3-8b-Instruct and Gemma-2-8b-it models. Although these models have been widely applied in existing works, their post-training procedure (SFT or RLHF) is not fully publicly disclosed. Additionally, while our experiments mainly focus on the widely applied DPO and SimPO objectives, a broader investigation with alternative models and preference learning methods is crucial in future work.

## Acknowledgments

This research is supported by the National Science and Technology Major Project (2023ZD0121102), the National Natural Science Foundation of China (U24B20180, 62121002). This research was also supported by the advanced computing resources provided by the Supercomputing Center of the USTC.

## Impact Statement

Our work introduces a data metric that enhances the alignment performance of large language models (LLMs) by selecting high-quality data for preference learning, potentially reducing misalignment risks in deployed AI systems. Our approach can also reduce computational resources during training by enabling more efficient training on high-quality data subsets, making responsible AI development more accessible. While our method improves data quality and efficiency, continued monitoring remains important to ensure AI systems are fair and beneficial.

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

# A. Related Work

**RLHF and Preference Learning Algorithms.** Despite RLHF's effectiveness in aligning language models with human preferences (Ziegler et al., 2019; Stiennon et al., 2020; Bai et al., 2022; Ouyang et al., 2022), the multi-stage RL training makes it computationally complex and hard to optimize (Santacroce et al., 2023; Zheng et al., 2023). Researchers have been exploring more efficient and simplified alignment algorithms, by simplifying the RL training process (Dong et al., 2023; Yuan et al., 2023) or utilizing only the offline preference dataset, with DPO (Rafailov et al., 2024) being a notable example. In addition to DPO, various offline preference learning objectives have been proposed, such as IPO (Azar et al., 2024), KTO (Ethayarajh et al., 2024), ORPO (Hong et al., 2024), and SimPO (Meng et al., 2024). Moreover, these offline preference optimization methods have been extended to iterative settings, with new preference pairs continuously sampled or reference policy updated using models trained in the previous iteration (Xu et al., 2023; Yuan et al., 2024; Rosset et al., 2024; Xiong et al., 2024).

**Data Quality in Alignment.** The importance of data quality in alignment processes has been well-documented, both in RLHF (Nakano et al., 2021; Bai et al., 2022) and offline preference learning methods. A significant body of research focuses on the distribution for response sampling during preference dataset construction (Liu et al., 2024a; Tajwar et al., 2024; Xiong et al., 2024), while another line of work examines the quality of different preference pairs within the preference dataset (Wu et al., 2024; 2025b; Morimura et al., 2024; Pattnaik et al., 2024; Wang et al., 2025). Of particular relevance to this study are efforts to explicitly define and utilize data quality metrics, such as leveraging the explicit reward margin to select high-quality data (Khaki et al., 2024; Ye et al., 2024) or reweighting loss (Wang et al., 2024b), and using the implicit reward margin to prioritize training data (Muldrew et al., 2024; Yang et al., 2024) or calibrating loss functions (Xu et al., 2023; Xiao et al., 2024). Despite the demonstrated effectiveness of these metrics, their often conflicting properties (as shown in Figure 1a) necessitate the development of a more universal data quality metric, which motivates us to propose the alignment potential metric.

Additionally, current works based on implicit reward margin are limited to iterative preference learning settings (Muldrew et al., 2024; Yang et al., 2024), requiring the model $\pi_\theta$ to be different from $\pi_{\mathrm{ref}}$ to avoid constant-zero implicit reward $\hat{r}_\theta$. To overcome such a shortcoming, we propose to utilize the SimPO-based implicit reward $\hat{r}_\theta^{\mathrm{Sim}}$, resulting in a new version of implicit reward margin applicable for standard offline preference learning scenarios.

**Self Play Alignment and Prompt Synthesis.** The paired comparison nature of preference learning has inspired a range of self-play methods based on two-player games (Munos et al., 2024; Chen et al., 2024; Rosset et al., 2024; Wu et al., 2025c), with both players being LMs generating responses to given prompts. Diverse from these works, eva (Ye et al., 2024) proposes an asymmetric alignment game involving a prompt creator and a response solver to augment preference data with higher reward differences (*aka.* informativeness in eva). To generate new prompts, researchers have developed various prompt synthesis methods such as SelfInstruct (Wang et al., 2023), EvolQuality (Liu et al., 2024b), EvolInstruct (Xu et al., 2024a), Magpie (Xu et al., 2024c) and so on[3]. Our work adopts the asymmetric self-play framework from eva to generate additional preference data, with a focus on evaluating various data quality metrics within this evolving data context.

# B. Additional Experiments

## B.1. Ablations study for Data Metrics Experiments

**Additional Reward Models.** For the ablation study concerning different reward models, SimPO provides an additional preference dataset generated using the Llama-3-8b-instruct model and annotated with the PairRM reward model, available at llama3-ultrafeedback. Given such convenience, we directly utilize this dataset and select top-40% subsets with different metrics for subsequent SimPO training. The performance of trained models with different metrics is depicted in Figure 8a. Consistent with the results discussed in § 3.2, our proposed alignment potential metric $M_{AP}$ still achieves the highest performance across all benchmarks under varying reward models, demonstrating the efficacy and robustness of our metrics.

**Additional Training Objectives.** For the ablation study focused on different training objectives, we opt for DPO loss as a substitute for SimPO and employ the Gemma-2-9b-it model, noted for its high performance. We again apply various metrics to select top-40% data for training. The results, depicted in Figure 8b, showcase that our proposed metric consistently outperforms existing baselines with notable improvements. These consistent performance gains certify the $M_{AP}$ metric's ability to identify high-quality data for alignment training, demonstrating stable improvements for varying

---

[3]Our contribution is orthogonal to different prompt synthesis techniques, and we employ EvolInstruct to generate new prompts following eva.

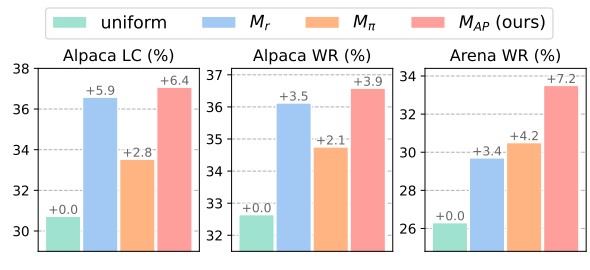
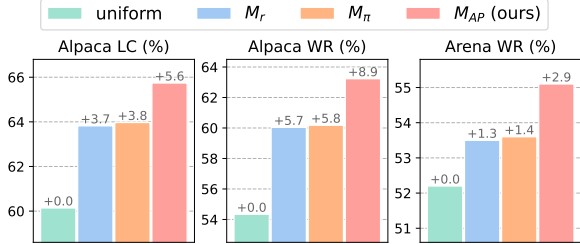

(a) Ablation on different reward models

(b) Ablation on different training objectives

*Figure 8.* **Ablation study for data metrics.** (8a) Performance of Llama-3-8b-instruct model trained with SimPO loss on preference pairs selected by different metrics. The preference dataset is based on the PariRM reward model, instead of the ArmoRM model considered in other settings. (8b) Performance of Gemma-2-9b-it model trained with DPO loss on preference pairs selected by different metrics.

training objectives. Additionally, we include IPO (Azar et al., 2024) as another ablation training objective, and include both PairRM and ArmoRM-based Llama datasets for experiments. Consistent with previous configurations, we select the top 40% of subsets based on various metrics for further training. As shown in Table 2, our proposed $M_{AP}$ metric still outperforms existing baselines under this alignment method, further certifying its effectiveness.

*Table 2.* Performance of Llama-3-8b-Instruct model trained with **IPO** on datasets selected by different metrics.

(a) Results on PairRM annotated dataset.

| Metrics | Uniform | $M_r$ | $M_\pi$ | $M_{AP}$ |
|---|---|---|---|---|
| **AE 2.0 LC** | 30.49 | 34.15 | 35.04 | **35.44** |
| **AE 2.0 WR** | 34.15 | 34.08 | 35.40 | **36.08** |

(b) Results on ArmoRM annotated dataset.

| Metrics | Uniform | $M_r$ | $M_\pi$ | $M_{AP}$ |
|---|---|---|---|---|
| **AE 2.0 LC** | 33.17 | 33.92 | 38.19 | **40.35** |
| **AE 2.0 WR** | 30.63 | 30.17 | 35.73 | **38.14** |

**Varying Data Amount.** In previous data selection experiments, we focused solely on a fixed proportion of the dataset subset, specifically 40%. In this ablation study, we explore the impact of varying the quantity of selected data. Utilizing the Llama-3-8b-Instruct model along with its PairRM-based dataset, we perform SimPO training with selection ratios of 20%, 40%, and 60%. Table 3 shows that our $M_{AP}$ metric consistently outperforms existing methods across different proportions of selected data.

*Table 3.* Performance of Llama-3-8b-Instruct model trained with SimPO on datasets selected with **varying selected ratio**.

| Ratio | Top-20% | | Top-40% | | Top-60% | |
|---|---|---|---|---|---|---|
| **AE 2.0** | **LC** | **WR** | **LC** | **WR** | **LC** | **WR** |
| Uniform | 26.02 | 28.06 | 30.72 | 32.64 | 33.94 | 34.72 |
| $M_r$ | 28.95 | 30.24 | 36.58 | 36.12 | 40.77 | 38.14 |
| $M_\pi$ | 28.46 | **31.01** | 33.53 | 34.75 | 38.29 | 37.28 |
| $M_{AP}$ | **30.66** | 30.62 | **37.07** | **36.58** | **42.86** | **39.60** |

**Additional Selection Strategies.** Beyond the existing $M_r$ and $M_\pi$ metrics, we broaden the data selection baselines by introducing two strategies: **(1) reversed data selection**: In this approach, we select the bottom-k data as ranked by various metrics. **(2) ranking-based metric**: Here, we propose selecting preference data where the ranking of implicit and explicit rewards for $y_w$ and $y_l$ is opposite. Specifically, this occurs when $\mathbb{I}(r(x, y_w) > r(x, y_l)) + \mathbb{I}(\hat{r}_\theta(x, y_w) > \hat{r}_\theta(x, y_l)) = 1$. This intuitive approach mirrors the $M_+$ metric as discussed in Section 3.1, both of which encourage the contradictions between the current LLM and the reward model's preference judgment. In contrast, our $M_{AP}$ incorporates a reward noise regularization design to avoid such contradictions, thus mitigating the reward noise problem.

For reversed data selection, we employ the Llama-3-8b-Instruct model and its ArmoRM-based dataset for SimPO training. As demonstrated in Table 4, the performance of resultant models across all metrics–when selecting the bottom 40% based on $M_r$, $-M_\pi$, $M_{AP}$–was poorer than the uniform baseline, indicating lower data quality

*Table 4.* Performance of Llama-3-8b-Instruct model trained with SimPO on datasets **reversely** selected with different metrics.

| Reversed Metrics | Uniform | $M_r$ | $M_\pi$ | $M_{AP}$ |
|---|---|---|---|---|
| **AE 2.0 LC** | 41.61 | 40.66 | 37.07 | *34.64* |
| **AE 2.0 WR** | 36.26 | 33.36 | 32.95 | *29.37* |

through reverse selection. Notably, the lowest performance observed was with our $M_{AP}$ metric, verifying its efficacy in identifying high-quality datasets from an opposite standpoint.

For the ranking-based selection, we select 40% subsets of Llama-based datasets where the ranking of explicit reward and implicit reward on $y_w$ and $y_l$ is opposite. As anticipated, this selection method clearly underperforms compared to the regularized $M_{AP}$ metric, reinforcing the justification of our reward noise regularization design.

*Table 5.* Additional comparisons with **ranking-based** selection. Experiments are conducted by training Llama-3-8b-Instruct model on different datasets using SimPO.



(a) Results on PairRM annotated dataset.

| Metrics | Uniform | Ranking | $M_{AP}$ |
|---|---|---|---|
| **AE 2.0 LC** | 30.72 | 31.53 | **37.07** |
| **AE 2.0 WR** | 32.64 | 30.40 | **36.58** |

(b) Results on ArmoRM annotated dataset.

| Metrics | Uniform | Ranking | $M_{AP}$ |
|---|---|---|---|
| **AE 2.0 LC** | 41.61 | 43.48 | **47.83** |
| **AE 2.0 WR** | 36.26 | 38.47 | **45.14** |



### B.2. Significance Testing

We present detailed results corresponding to the figures in the main body of this paper (i.e., Figure 4 and Figure 5), including the standard deviation of AlpacaEval 2 and the 95% confidence interval of Arena Hard. As shown in Table 6, all benchmark results are reasonable and do not display significant outliers or noise.

### B.3. More Comparison with EVA

---
**Algorithm 2** Eva with *Select-then-Evolve* Pipeline
---
**Input:** SFT model $\pi_{\text{SFT}}$, reward model $r$, existing prompts $\mathcal{X}$ and data quality metric $M$.
Initialize $\pi_{\theta_0}$ with $\pi_{\text{SFT}}$.
**for** iteration $t = 1, \ldots, T$ **do**
    Generate initial preference dataset: $\mathcal{D} \leftarrow \text{GenDataset}(\mathcal{X}, \pi_{\theta_{t-1}}, r)$.
    Select top-$k$ dataset using $M$ metric:
        $\mathcal{D}_k = \{(x, y_w, y_l) \in \mathcal{D} | M(x, y_w, y_l; \theta_{t-1}) \geq \tau_k\}$, with $\tau_k$ being the $k$-th largest $M$ score in $\mathcal{D}$.
    Evolve prompts: $\mathcal{X}'_k \leftarrow \text{EvolInstruct}(\mathcal{X}_k)$, where $\mathcal{X}_k$ denotes the prompts of $\mathcal{D}_k$ dataset.
    Generate additional preference dataset: $\mathcal{D}'_k \leftarrow \text{GenDataset}(\mathcal{X}'_k, \pi_{\theta_{t-1}}, r)$.
    Conduct preference optimization on $\mathcal{D} \cup \mathcal{D}'_k$: $\theta_t \leftarrow \theta_{t-1} - \eta \nabla_\theta \mathcal{L}(\theta_{t-1}; \mathcal{D} \cup \mathcal{D}'_k)$.
**end for**
**Return:** optimized policy $\pi_{\theta_T}$

---

Our main study employs the *evolve-then-select* pipeline for preference data generation, as detailed in Algorithm 1, where we streamline the dataset generation process using the $\text{GenDataset}(\mathcal{X}, \pi_\theta, r)$ function. This function takes the prompts $\mathcal{X}$, samples responses with LLM $\pi_\theta$, and annotates preferences with reward model $r$ to construct a preference dataset. The *evolve-then-select* pipeline ensures the final dataset $\mathcal{D}'_k$ aligns with the evaluation of data quality metrics.

In contrast, the original eva framework operates under a *select-then-evolve* pipeline[4], as detailed in Algorithm 2. Within this pipeline, data selection precedes the evolving data process. Therefore the data quality of the final evolved dataset $\mathcal{D}'_k$ may not be consistent with that of $\mathcal{D}_k$ due to the intricate prompt evolving and preference data generation process in data evolving. Nevertheless, we still include comparisons under this *select-then-evolve* setting to ensure a comprehensive and equitable evaluation against eva.

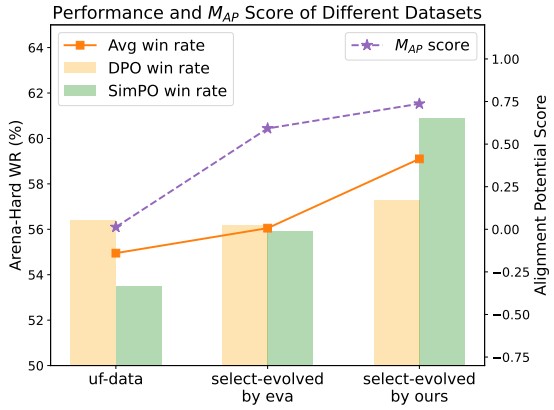

*Figure 9.* The $M_{AP}$ scores of different datasets and the corresponding Gemma models' performance. The dataset select-evolved by our methods shows the highest alignment potential scores and produces the best-performing results.

---
[4]The original eva paper employs $M_r$ scores as sampling weights for $\mathcal{D}_k$ selection but lacks implementation guidance. Thus we substitute this sampling with a similar top-$k$ selection.

*Table 6.* Standard deviation and 95% confidence intervals of benchmark results.

(a) Results of Llama-3-8b-Instruct model (Figure 4)

| Metrics | Uniform | $M_r$ | $M_\pi$ | $M_{AP}$ |
|---|---|---|---|---|
| AE 2.0 LC | 41.61 | 44.60 | 47.15 | **47.83** |
| AE 2.0 WR | 36.26 | 40.53 | 41.22 | **45.14** |
| AE 2.0 std | 1.45 | 1.44 | 1.43 | 1.46 |
| Arena WR | 29.9 | 33.5 | 33.0 | **34.6** |
| Arena CI | (-2.1, 1.7) | (-2.6, 2.4) | (-2.3, 2.0) | (-2.0, 2.4) |

(b) Results of Gemma-2-9b-it model (Figure 5)

| Metrics | Uniform | $M_r$ | $M_\pi$ | $M_{AP}$ |
|---|---|---|---|---|
| AE 2.0 LC | 67.95 | 68.63 | 68.25 | **70.45** |
| AE 2.0 WR | 61.40 | 68.64 | 69.45 | **70.45** |
| AE 2.0 std | 1.45 | 1.39 | 1.38 | 1.40 |
| Arena WR | 55.7 | 59.6 | 61.9 | **62.8** |
| Arena CI | (-2.6, 2.3) | (-2.7, 3.0) | (-2.7, 2.3) | (-2.3, 1.8) |

Using the experimental setup detailed in §5.1, but reversing the selection and evolution order, we assess models trained on various datasets as reported in Table 7. The results indicate that datasets select-evolved with our metric continue to deliver better overall performance than the supplementary UltraFeedback datasets and those derived using the eva method. To corroborate that these performance improvements stem from the ability to select high-quality data, we present the average performance (Arena Hard win rate) of Gemma models trained with either DPO or SimPO, alongside the alignment potential scores of the respective datasets, in Figure 9. As illustrated, the model performance continually improves as the dataset's alignment potential score increases, with the dataset select-evolved by our metric exhibiting both the highest alignment potential score and performance. This correlation between alignment potential score and performance further underscores our metric's capability of discerning high-quality data for training.

*Table 7.* Main experiments under the select-then-evolve self-play setting

| Method | Llama-3-8b-instruct | | | Gemma-2-9b-it | | |
|---|---|---|---|---|---|---|
| | AH | AE 2.0 | | AH | AE 2.0 | |
| | WR | LC | WR | WR | LC | WR |
| SFT | 21.4 | 23.25 | 23.50 | 40.7 | 48.75 | 36.49 |
| DPO$_{\text{uf-10k}}$ | 23.6 | 29.80 | 28.32 | 46.0 | 54.00 | 46.11 |
| +uf-10k | 35.0 | **42.67** | 41.67 | 56.4 | **62.56** | 62.17 |
| +eva-10k | 35.0 | 40.38 | 42.10 | 56.2 | 61.73 | **62.94** |
| **+ours-10k** | **35.5** | 41.98 | **44.29** | **57.3** | 62.07 | 62.42 |
| SimPO$_{\text{uf-10k}}$ | 24.2 | 28.45 | 26.23 | 45.2 | 56.90 | 43.80 |
| +uf-10k | 28.5 | 35.75 | 32.82 | 53.5 | **66.43** | 64.55 |
| +eva-10k | 32.7 | 40.78 | 38.68 | 55.9 | 64.96 | 62.88 |
| **+ours-10k** | **33.3** | **42.75** | **41.72** | **60.9** | 65.29 | **65.47** |

# C. Convergence Rates of DPO with Different Samplers

### C.1. Proof of Theorem 3.1

Here we restate the DPO loss in its population form:

$$
\begin{aligned}
\mathbb{E}_{(x,y,y')\sim s}\left[\mathcal{L}(x,y,y';\theta)\right] &= \mathbb{E}_{(x,y,y')\sim s}\left[-p(y \succ y')\log\sigma(\hat{r}_\theta(y) - \hat{r}_\theta(y'))\right] \\
&= \mathbb{E}_{(x,y,y')\sim s}\left[-\sigma(r(x,y) - r(x,y'))\log\sigma(\hat{r}_\theta(y) - \hat{r}_\theta(y'))\right].
\end{aligned}
$$

Under stochastic gradient descent, a data $(x^t, y^t, y'^t)$ is sampled to update the parameter $\theta^t$ at each iteration $t$ with the empirical loss. Since the sampled context $x^t$ and arms $y^t, y'^t$ can be utilized to compute both $\mathcal{L}(x^t, y^t, y'^t; \theta^t)$ and $\mathcal{L}(x^t, y'^t, y^t; \theta^t)$, we utilize both terms to derive the stochastic update rule, like previous works (Azar et al., 2024; Rosset et al., 2024):

$$
\theta^{t+1} = \theta^t - \frac{1}{2}\eta^t \nabla_\theta\left[\mathcal{L}(x^t, y^t, y'^t; \theta^t) + \mathcal{L}(x^t, y'^t, y^t; \theta^t)\right] \tag{12}
$$

where $\eta^t$ denotes the time-varying learning rate. Without loss of generality, we assume a uniform reference policy $\pi_{\text{ref}}(y|x) = \frac{1}{|\mathcal{Y}|}$ like Shi et al. (2025). Therefore $\hat{r}_\theta(x,y) = \beta \log \frac{\pi_\theta(y|x)}{1/|\mathcal{Y}|} = \beta\theta(x,y) + \log Z(x)$ and the optimum $\theta^*$, characterized by $r(x,y) = \hat{r}_{\theta^*}(x,y) + \log Z^*(x)$, turns into $\beta\theta^*(x,y) = r(x,y) + c(x)$, where $c(x)$ is a constant independent of $y$. The proof of Theorem 3.1 is as follows:

*Proof of Theorem 3.1.* For ease of annotation, we first consider the single context setting $|\mathcal{X}| = 1$ so that we can denote $\theta_y = \theta(x,y), r_y = r(x,y)$ with $x$ omitted for simplicity. We first investigate the stochastic update rule of $\theta^t$ given by Equation (12). Let the arms chosen at iteration $t$ being $y, y'$, we have:

$$
\begin{aligned}
\theta_y^{t+1} &= \theta_y^t + \frac{\eta^t\beta}{2}\left[\sigma(r_y - r_{y'})\sigma(\beta\theta_{y'}^t - \beta\theta_y^t) - \sigma(r_{y'} - r_y)\sigma(\beta\theta_y^t - \beta\theta_{y'}^t)\right] \\
&= \theta_y^t + \frac{\eta^t\beta}{2}\left[\sigma(r_y - r_{y'}) - \sigma(\beta\theta_y^t - \beta\theta_{y'}^t)\right] \\
\theta_{y'}^{t+1} &= \theta_{y'}^t + \frac{\eta^t\beta}{2}\left[\sigma(r_{y'} - r_y) - \sigma(\beta\theta_{y'}^t - \beta\theta_y^t)\right] \\
&= \theta_{y'}^t - \frac{\eta^t\beta}{2}\left[\sigma(r_y - r_{y'}) - \sigma(\beta\theta_y^t - \beta\theta_{y'}^t)\right],
\end{aligned}
$$

where the second equality holds by the fact that $\sigma(x)\sigma(-y) - \sigma(-x)\sigma(y) = \sigma(x) - \sigma(y)$.

Since the optimal target is $\beta\theta^* = r + c$, where $c$ is a constant, we denote a new vector $\xi^t = \beta\theta^t - r$. Let:

$$\Delta_{yy'}^t = \sigma(r_y - r_{y'}) - \sigma(\beta\theta_y^t - \beta\theta_{y'}^t).$$

The update rule can be rewritten as:

$$
\begin{aligned}
\theta_y^{t+1} &= \theta_y^t + \frac{\eta^t\beta}{2}\Delta_{yy'}^t, \theta_{y'}^{t+1} = \theta_{y'}^t - \frac{\eta^t\beta}{2}\Delta_{yy'}^t. \\
\xi_y^{t+1} &= \xi_y^t + \frac{\eta^t\beta^2}{2}\Delta_{yy'}^t, \xi_{y'}^{t+1} = \xi_{y'}^t - \frac{\eta^t\beta^2}{2}\Delta_{yy'}^t.
\end{aligned}
\tag{13}
$$

Note that $\theta_y^{t+1} + \theta_{y'}^{t+1} = \theta_y^t + \theta_{y'}^t$ and $\xi_y^{t+1} + \xi_{y'}^{t+1} = \xi_y^t + \xi_{y'}^t$, which indicates the mean of $\theta$ and $\xi$ remains unchanged during optimization.

For optimal $\theta^*$, the $\xi^* = \beta\theta^* - r$ should be a constant vector $c \cdot \mathbf{1}$. Therefore, we can measure the variance of $\xi^t$ as an indicator of convergence:

$$V^t = \frac{1}{|\mathcal{Y}|}\sum_{y\in\mathcal{Y}}(\xi_y^t - \bar{\xi}^t)^2, \text{ where } \bar{\xi}^t = \frac{1}{|\mathcal{Y}|}\sum_{y\in\mathcal{Y}}\xi_y^t.$$

Using $V^t$, the distance function $\text{Dist}(\theta^t, \theta^*)$ satisfies that:

$$\text{Dist}(\theta^t, \theta^*)^2 = \frac{1}{|\mathcal{Y}|^2}\sum_{y,y'\in\mathcal{Y}}M_1(x,y,y'|\theta,r)^2 = \frac{1}{|\mathcal{Y}|^2}\sum_{y,y'\in\mathcal{Y}}(\xi_y^t - \xi_{y'}^t)^2 = 2V^t.$$

Now we examine how $V^t$ changes at each iteration. Assume the sampled pairs are $y, y'$ at iteration $t$, then we have:

$$
\begin{aligned}
V^{t+1} &= V^t + \frac{1}{|\mathcal{Y}|}\left[(\xi_y^{t+1} - \bar{\xi}^{t+1})^2 + (\xi_{y'}^{t+1} - \bar{\xi}^{t+1})^2 - (\xi_y^t - \bar{\xi}^t)^2 - (\xi_{y'}^t - \bar{\xi}^t)^2\right] \\
&= V^t + \frac{1}{|\mathcal{Y}|}\left[(\xi_y^{t+1})^2 + (\xi_{y'}^{t+1})^2 - (\xi_y^t)^2 - (\xi_{y'}^t)^2\right] \\
&= V^t + \frac{1}{|\mathcal{Y}|}\left[(\xi_y^t + a\Delta_{yy'}^t)^2 + (\xi_{y'}^t - a\Delta_{yy'}^t)^2 - (\xi_y^t)^2 - (\xi_{y'}^t)^2\right] \quad (\text{let } a = \frac{\eta^t\beta^2}{2}) \\
&= V^t + \frac{2a}{|\mathcal{Y}|}(\xi_y^t - \xi_{y'}^t + a\Delta_{yy'}^t)\Delta_{yy'}^t.
\end{aligned}
$$

where the second equation holds with $\bar{\xi}^{t+1} = \bar{\xi}^t$ and $\xi_y^{t+1} + \xi_{y'}^{t+1} = \xi_y^t + \xi_{y'}^t$.

Using the mean value theorem, we have:

$$\Delta_{yy'}^t = \sigma'(\lambda_{yy'}^t)\left[(r_y - r_{y'}) - (\beta\theta_y - \beta\theta_{y'})\right] = -\sigma'(\lambda_{yy'}^t)(\xi_y^t - \xi_{y'}^t),$$

where $\lambda_{yy'}^t$ is between $r_y - r_{y'}$ and $\beta\theta_y^t - \beta\theta_{y'}^t$. Note that $\sigma'(\cdot) \in (0, \frac{1}{4}]$, and the value of $\sigma'(\lambda)$ can be further lower bounded using the update rule.

The change in $V^t$ can thus be written as:

$$\Delta V^t = V^{t+1} - V^t = -\frac{2(1 - a\sigma'(\lambda_{yy'}^t))a\sigma'(\lambda_{yy'}^t)}{|\mathcal{Y}|}(\xi_y^t - \xi_{y'}^t)^2$$

Under the optimal learning rate $\eta^t = \frac{1}{\beta^2\sigma'(\lambda_{yy'}^t)}$ (we can adjust the learning rate at each iteration, similar to the line search methods), such that $a\sigma'(\lambda_{yy'}^t) = \frac{1}{2}$, the change in $V^t$ satisfies:

$$\Delta V^t = -\frac{1}{2|\mathcal{Y}|}(\xi_y^t - \xi_{y'}^t)^2. \tag{14}$$

Now we can examine the impact of different samplers, which determine the choices of $y, y'$ at each iteration. For the uniform sampler, under the optimal learning rate $\eta^t$, we have:

$$\begin{aligned}
\mathbb{E}[V^{t+1}|V^t] &= V^t + \mathbb{E}[\Delta V^t] \\
&= V^t - \frac{1}{2|\mathcal{Y}|}\mathbb{E}_{y,y'\sim s_u}[(\xi_y^t - \xi_{y'}^t)^2] \\
&= V^t - \frac{1}{2|\mathcal{Y}|} \cdot \frac{1}{|\mathcal{Y}|^2}\sum_{y\in\mathcal{Y}}\sum_{y'\in\mathcal{Y}}(\xi_y^t - \xi_{y'}^t)^2 \\
&= V^t - \frac{1}{2|\mathcal{Y}|} \cdot 2V^t \\
&= (1 - \frac{1}{|\mathcal{Y}|})V^t.
\end{aligned}$$

So we have the following **convergence result for the uniform sampler**:

$$\mathbb{E}[V^t] = (1 - \frac{1}{|\mathcal{Y}|})^t V^0. \tag{15}$$

In contrast, the adversarial sampler, which chooses the pair $y^*, y'^*$ to maximize the $M_1$ score:

$$\begin{aligned}
y*, y'^* &= \arg\max_{y,y'\in\mathcal{Y}} M_1(x, y, y') \\
&= \arg\max_{y,y'\in\mathcal{Y}} |r_y - r_{y'} - (\beta\theta_y^t - \beta\theta_{y'}^t)| \\
&= \arg\max_{y,y'\in\mathcal{Y}} |\xi_y^t - \xi_{y'}^t|,
\end{aligned}$$

which means to select the maximal and minimal elements from $\xi^t$. Given this definition, the selected pairs at each iteration $y^*, y'^*$ satisfy that:

$$(\xi_{y^*}^t - \xi_y^t) \cdot (\xi_{y'^*}^t - \xi_y^t) \le 0, \forall y \in \mathcal{Y}.$$

Let $m^t = \xi_{y^*}^t - \bar{\xi}^t, M^t = \xi_{y'^*}^t - \bar{\xi}^t$ and $s_y^t = \xi_y^t - \bar{\xi}^t$, the inequality above can be rewritten as:

$$(m^t - s_y^t)(M^t - s_y^t) \le 0, \forall y \in \mathcal{Y}.$$

Summarize over all $y \in \mathcal{Y}$, we have:

$$
\begin{aligned}
0 &\geq \sum_{y \in \mathcal{Y}} (m^t - s_y^t)(M^t - s_y^t) \\
&= |\mathcal{Y}| \cdot m^t M^t - (m^t + M^t) \sum_{y \in \mathcal{Y}} s_y^t + \sum_{y \in \mathcal{Y}} (s_y^t)^2 \\
&= |\mathcal{Y}| \cdot m^t M^t + \sum_{y \in \mathcal{Y}} (s_y^t)^2 \\
&= |\mathcal{Y}| \cdot \left[ (\xi_{y^*}^t - \bar{\xi}^t)(\xi_{y'^*}^t - \bar{\xi}^t) + V^t \right].
\end{aligned}
$$

where the second equality (line 2 to line 3) holds by $\sum_{y \in \mathcal{Y}} s_y^t = 0$. Combing with the fact that $(a - b)^2 \geq -4ab$, we have:

$$
\begin{aligned}
(\xi_{y^*}^t - \xi_{y'^*}^t)^2 &\geq -4(\xi_{y^*}^t - \bar{\xi}^t)(\xi_{y'^*}^t - \bar{\xi}^t) \\
&\geq 4V^t.
\end{aligned}
$$

Therefore, when updating using the selected pair $y^*, y'^*$ at iteration $t$, we have (under the optimal learning rate in Equation (14)):

$$
\begin{aligned}
V^{t+1} &= V^t - \frac{1}{2|\mathcal{Y}|}(\xi_{y^*}^t - \xi_{y'^*}^t)^2 \\
&\leq V^t - \frac{2}{|\mathcal{Y}|} V^t \\
&= (1 - \frac{2}{|\mathcal{Y}|}) V^t.
\end{aligned}
$$

So we have the following **convergence result for the adversarial sampler**:

$$
V^t \leq (1 - \frac{2}{|\mathcal{Y}|})^t V^0, \tag{16}
$$

Eventually, we can compare the iterations required to converge to the same level of error, when applying different samplers. Since $\text{Dist}(\theta^t, \theta^*)^2 = 2V^t$, to reach $\text{Dist}(\theta^t, \theta^*) = \varepsilon \text{Dist}(\theta^0, \theta^*)$ for some small $\varepsilon < 1$, or equivalently, $V^t = \varepsilon^2 V^0$, the uniform sampler would require $T_u(\varepsilon) = \frac{2\log(\varepsilon)}{\log(1 - 1/|\mathcal{Y}|)}$ iterations according to Equation (15). In comparison, the adversarial sampler would require $T_{adv}(\varepsilon) = \frac{2\log(\varepsilon)}{\log(1 - 2/|\mathcal{Y}|)}$ iterations by Equation (16). As one may verify that for $|\mathcal{Y}| > 2$:

$$
\log(1 - \frac{2}{|\mathcal{Y}|}) > 2\log(1 - \frac{1}{|\mathcal{Y}|}),
$$

therefore **the adversarial sampler requires less than half the iterations of the uniform counterpart**:

$$
T_{adv}(\varepsilon) < 0.5 T_u(\varepsilon). \tag{17}
$$

Furthermore, the above results can directly generalize to the contextual setting where $|\mathcal{X}| > 1$, which we briefly discuss here: With slight abuse of annotations, we can re-define $V^t = \frac{1}{|\mathcal{X}||\mathcal{Y}|} \sum_{x \in \mathcal{X}} \sum_{y \in \mathcal{Y}} (\xi_{x,y}^t - \bar{\xi}_x^t)^2$, and the update of $V^t$ with preference data $(x, y, y')$ at $t$ would be:

$$
V^{t+1} = V^t + \frac{2a}{|\mathcal{X}||\mathcal{Y}|}(\xi_{xy}^t - \xi_{xy'}^t + a\Delta_{xyy'}^t)\Delta_{xyy'}^t.
$$

Then the convergence results immediately become $\mathbb{E}[V^t] = (1 - \frac{1}{|\mathcal{X}||\mathcal{Y}|})^t V^0$ for uniform sampler and $V^t \leq (1 - \frac{2}{|\mathcal{X}||\mathcal{Y}|}) V^0$ for the adversarial sampler, leading to the same result with Equation (17). $\square$

## C.2. Numerical Experiments

Theorem 3.1 establishes that the adversarial sampler requires less than half the iterations of the uniform sampler to reach the same error level. Here we conduct numerical experiments to validate this result.

We examine two scenarios: the standard bandit setting with a single context (*i.e.*, $|\mathcal{X}| = 1$) and a contextual setting with $|\mathcal{X}| = 5$. In both scenarios, the arm space is set to $|\mathcal{Y}| = 10$. Rewards are initialized uniformly using $U[0, 1]$ and the initial parameter $\theta_x^0$ is set to 0. We consider a uniform reference policy, with the KL-parameter $\beta$ fixed at 0.1. For the learning rate, while our proof uses the optimal learning rate $\eta^t = \frac{1}{\beta^2 \sigma'(\lambda_{yy'}^t)} \geq \frac{4}{\beta^2}$, a fixed learning rate is set to $4/\beta^2$ for simplicity.

Figure 10 presents the error, *i.e.*, the distance to optimum $\mathrm{Dist}(\theta, \theta^*)$, averaged across 10 random initializations. As depicted, while both samplers achieve linear convergence, the adversarial sampler converges significantly more rapidly due to its strategic selection of the data with the most potential for alignment. Notably, in our experiments, the uniform sampler required approximately six times more iterations to reach the same level of error as the adversarial sampler, which corroborates the theoretical prediction of a speed-up by a factor of more than two.

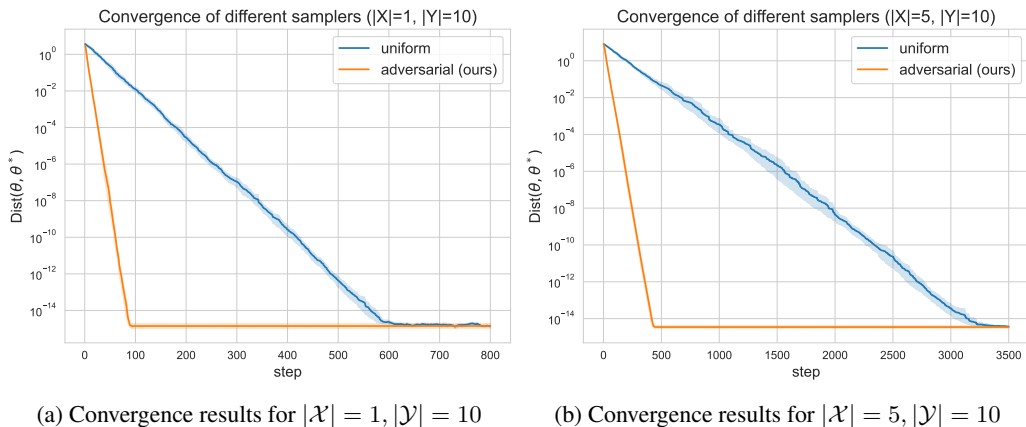

(a) Convergence results for $|\mathcal{X}| = 1, |\mathcal{Y}| = 10$       (b) Convergence results for $|\mathcal{X}| = 5, |\mathcal{Y}| = 10$

*Figure 10.* **Numerical experiments for DPO under contextual bandit setting.** This figure compares learning convergence using two samplers: a uniform sampler and an adversarial sampler that selects pair data to maximize our $M_1$ metric. The plots display the error $\mathrm{Dist}(\theta, \theta^*)$ across 10 random initializations. Figure 10a addresses the single context scenario, while Figure 10b pertains to multiple contexts. The error lower bound of approximately 1e-15 is attributed to floating-point precision limitations.

# D. Implementation Details

## D.1. Codes & Hyperparameters

Our code is open-sourced in: https://github.com/Hesse73/Alignment-Potential-Metric.

**Data quality metrics.** We mainly investigate three metrics in this paper: (1) the explicit reward margin metric: $M_r(x, y_w, y_l) = |r(x, y_w) - r(x, y_l)|$; (2) the implicit reward margin metric: $M_\pi(x, y_w, y_l) = |\hat{r}_\theta(x, y_w) - \hat{r}_\theta(x, y_l)|$; and (3) our proposed alignment potential metric: $M_{AP} = |r(x, y_w) - r(x, y_l)| - |\hat{r}_\theta(x, y_w) - \hat{r}_\theta(x, y_l)|$. While explicit rewards $r(x, y)$ are directly given by a reward model, computing the implicit rewards $\hat{r}_\theta(x, y) = \beta \log \frac{\pi_\theta(y|x)}{\pi_{\mathrm{ref}}(y|x)}$ requires the LLM $\pi_\theta$, a reference policy $\pi_{\mathrm{ref}}$, and a parameter $\beta > 0$. As discussed stated in §3.2, since the implicit reward $\hat{r}_\theta(x, y)$ becomes constant-zero when $\pi_\theta$ is identical with $\pi_{\mathrm{ref}}$, we propose implementing the implicit reward via SimPO's length-normalized reward: $\hat{r}_\theta^{\mathrm{Sim}}(x, y) = \beta \log \pi_\theta(y|x)/|y|$, which can be seen as a special case of $\hat{r}_\theta$ with uniform reference model (Wu et al., 2025a). Regarding the parameter $\beta$, while it doesn't influence the ranking determined by $M_\pi$, it does affect the score produced by our $M_{AP}$ metric. To effectively choose a value for $\beta$, let $\sigma_r$ and $\sigma_\pi$ denote the standard deviations of $|r(x, y_w) - r(x, y_l)|$ and $|\log \pi_\theta(y_w|x)/|y_w| - \log \pi_\theta(y_l|x)/|y_l||$ across the dataset, respectively. Utilizing these standard deviations, we rescale the two margins akin to Z-score normalization and introduce a parameter $\alpha \in \{0.25, 0.5, 1.0, 2.5, 5.0\}$ for computing the $M_{AP}$ metric:

$$M_{AP}(x, y_w, y_l) = \frac{1}{\sigma_r} |r(x, y_w) - r(x, y_l)| - \frac{\alpha}{\sigma_\pi} \left| \frac{\log \pi_\theta(y_w|x)}{|y_w|} - \frac{\log \pi_\theta(y_l|x)}{|y_l|} \right|.$$

Here we use $\alpha$ to distinguish it from the $\beta$ hyperparameter used in the DPO and SimPO loss function.

**Preliminary experiments.** The preliminary experiments, outlined in Section 3.2, employ the three metrics to select the top 40% subsets from existing preference datasets within SimPO. While the explicit rewards $r(x, y)$ are present in SimPO's datasets, they didn't record the probabilities of LLM $\pi_\theta(y|x)$ when sampling the responses. So we use the corresponding models from huggingface: Gemma-2-9b-it and Llama-3-8b-instruct, input the gener-ated responses $y$ and prompts $y$, and compute $\pi_\theta(y|x)$ to

*Table 8.* Hyperparameters used in our preliminary experiments.

| Hyperparameter | Gemma (ArmoRM) | | Llama (SimPO) | |
|---|---|---|---|---|
| | SimPO | DPO | ArmoRM | PairRM |
| learning rate | 8e-7 | 5e-7 | 1e-6 | 1e-6 |
| $\beta$ | 10 | 0.01 | 10 | 2.5 |
| $\gamma/\beta$ | 0.5 | \ | 0.3 | 0.55 |

derive implicit rewards $\log \pi_\theta(y|x)/|y|$. We use three datasets for data selection: two Llama-based datasets annotated by ArmoRM: llama3-ultrafeedback-armorm and PairRM: llama3-ultrafeedback, and one Gemma-based dataset annotated by ArmoRM: gemma2-ultrafeedback-armorm. For the hyperparameter in $M_{AP}$, we set $\alpha = 1.0$ for Llama's PairRM annotated dataset and set $\alpha = 2.5$ for the remaining two datasets. After data selection, we apply the tuned hyperparameters reported in SimPO's paper for both SimPO and DPO training. Specifically, we set a batch size of 128, with a max sequence length of 2048 and a max prompt length of 1800. All the models are trained using a cosine learning rate scheduler with a 10% warmup ratio, via AdamW optimizer for one epoch. Table 8 details the other hyperparameters for different datasets and objectives

**Main experiments.** In the main experiments, we integrate our metric $M_{AP}$ into the evolving alignment (eva) framework, which involves iterative preference dataset generation and prompt evolving processes. To generate the preference datasets $\mathcal{D}$ and $\mathcal{D}'$, we use top-p sampling (p=0.95) with a temperature of 0.8 and max sequence length of 4096 to sample responses $y$ for prompts $x$. Following SimPO, we use five distinct random seeds to sample 5 responses $\{y_1, \ldots, y_5\}$ for each prompt $x$, and use the responses with max/min reward $r(x, y)$, annotated via ArmoRM, to construct the preference pair $(x, y_w, y_l)$. The previous sampling parameters are also used for prompt evolving, *i.e.,* query the model to rewrite existing prompts $x \in \mathcal{X}$ into evolved prompts $\mathcal{X}'$. Regarding the instruction for LLMs to write prompts, we employ the five prompts in EvolInstruct (Xu et al., 2024a) and make some modifications for better instruction-following capabilities. Detailed descriptions of these revised prompts are available in Table 9. Incorrectly formatted prompts per the evolving instruction are filtered out when constructing $\mathcal{X}'$.

As for the subsequent data selection and training, we use $\alpha = 2.5$ for gemma and $\alpha = 5.0$ for llama for our main results (Table 1 and Table 7). For llama with DPO, we set a learning rate of 7e-7 and $\beta = 0.01$, and other hyperparameters remain consistent with Table 8. In the data-size scaling experiment (Figure 6), the number of EvolInstruct prompt templates is reduced from 5 to 2, ensuring the processed prompt set is less than 30k×2, maintaining a smaller prompt set than 60k. The hyperparameter $\alpha = 0.5$ and $\gamma/\beta = 0.3$ is employed for $M_{AP}$ metric and subsequent SimPO training. For the multi-iteration experiment (Figure 7), we reduce the 3rd iteration's learning rate to 3e-7 to prevent overfitting. The $\gamma/\beta$ parameter is tuned within $\{0.1, 0.3, 0.5, 0.8\}$, and the best-performing results based on the Arena Hard WR are reported.

### D.2. GPT-4 Annotation

In §3.1, we use $M_+$ and $M_{AP}$ metrics to select the top-10% subsets from SimPO's preference dataset and measure GPT-4's agreement on the preference annotation in data $(x, y_w, y_l)$. Specifically, we input the prompt $x$ along with two responses $y_w, y_l$ for GPT-4, ask it to choose the better one, and check if it aligns with the preference data's judgment, *i.e.,* $y_w \succ y_l|x$. For feasibility and financial consideration, 1,000 preference data from the selected top subsets is sampled for annotation. We employ the prompt template from Arena-hard (Li et al., 2024) to prompt the GPT-4-1106-preview model for preference annotation. The prompts can be accessed via their config file.

In addition to the agreement results in Figure 3, which is selected and annotated on the Gemma dataset of SimPO, we also include the agreement results of the Llama dataset of SimPO in Figure 11. Here data selected by $M_{AP}$ metric also results in a higher agree rate than $M_+$, underscoring the effectiveness of noise reduction strategies in $M_{AP}$.

## E. Illustrative Examples of Preference Data

This section provides detailed examples of preference data utilized in this paper.

**Contradiction of existing metrics.** Prior research indicates that prioritizing data with *large explicit reward margins* or *smaller implicit reward margins* can improve alignment performance. However, these two metrics often give conflicting

GPT-4 Agreement with Reward Model on Selected Data

*Figure 11.* GPT-4 agreement with reward model on data selected by different metrics from SimPO's dataset with Llama model. The data selected by $M_{AP}$ has a higher agree rate than $M_+$, indicating less reward annotation noise.

guidance, demonstrated by two examples from the SimPO preference dataset (gemma2-ultrafeedback-armorm) depicted in Figure 1a: The first example, where the data produces both large explicit and implicit reward margins, is rated as "high-quality" by the explicit reward margin but "low-quality" by the implicit reward margin. Detailed information for this example is provided in Table 10. The second example, where the data produces both small explicit and implicit reward margins, is rated as "low-quality" by the explicit reward margin but "high-quality" by the implicit reward margin. This data is elucidated in Table 11.

Our proposed alignment potential metric differentiates itself by classifying both examples as "low-quality." This is due to the small disparity between explicit and implicit rewards, indicating that the LLM's preferences are already well-aligned with the data, thus negating the need for further training.

**Noisy reward model annotation.** Reward models can introduce noisy preferences in annotation. An illustrative example from the gemma2-ultrafeedback-armorm dataset is presented in Table 12, with its simplified version illustrated in Figure 2 in the main body of our paper.

As shown in the example, the prompt asks LLM about the name of Zulu soldiers and requires it to output "No answer" if the answer cannot be determined by the provided text. Since the provided text does not state what Zulu soldiers are called, the correct answer would be "No answer", corresponding to the $y_l$ response. However, the reward model predicts a lower reward for the correct answer and incorrectly assigns a higher reward for the wrong answer $y_w$. This highlights the limitations of reward models and the resultant annotation noise.

Moreover, this preference data also implies that the implicit reward, given by the LLM itself, could provide correct preference annotations: the implicit reward $\hat{r}_\theta(x, y_l) = -3.4$ for the correct yet mislabeled response $y_l$, is much higher than the wrong answer $\hat{r}_\theta(x, y_w) = -8.9$. This observation is supported by recent studies recognizing the LLM's capacity to annotate preferences (Chen et al., 2025; Xu et al., 2024b).

*Table 9.* Prompts to evolve instructions, adapted from EvolInstruct (Xu et al., 2024a).

| **In-Depth Base Prompt** with an adaptive "strategy" identifier: `<STRATEGY>`, which will be replaced by the following 4 strategies | `I want you act as a Prompt Rewriter.\n Your objective is to rewrite a {Given Prompt} into a **more complex version** to make those famous AI systems (e.g., ChatGPT)a bit harder to handle.\n **Requirements**:\n - The {Rewritten Prompt} cannot omit the input in the {Given Prompt}. \n - You SHOULD complicate the given prompt using the following method: \n <STRATEGY> \n - The {Rewritten Prompt} must be reasonable and must be understood and responded to by humans.\n **Constraints that must be followed**:\n - The {Rewritten Prompt} can only add 10 to 20 words into the {Given Prompt}. \n - The {Rewritten Prompt} should be self-contained, with **all necessary information** provided, so that it can be responded to without needing to refer back to the {Given Prompt}.\n - Your response should contain **only** the {Rewritten Prompt}, **without any** additional formatting or introductory phrases such as 'Here is the rewritten prompt:' or 'The rewritten prompt is:'.\n The {Given Prompt}: \n<PROMPT>\n ========\nBased on the prompt above, rewrite a prompt:\n{Rewritten Prompt}:\n` |
|---|---|
| **– Adding Constraints** | The strategy is: `Please add one more constraint/requirements into the {Given Prompt}` |
| **– Deppening** | The strategy is: `If the {Given Prompt} contains inquiries about certain issues, the depth and breadth of the inquiry can be increased.` |
| **– Concretizing** | The strategy is: `Please replace general concepts with more specific concepts.` |
| **– Increasing Reasoning** | The strategy is: `If the {Given Prompt} can be solved with just a few simple thinking processes, you can rewrite it to explicitly request multiple-step reasoning.` |
| **In-Breadth Prompt** | `I want you to act as a Prompt Creator.\n Your objective is to take inspiration from the {Given Prompt} to create **one** brand new prompt.\n **Reqiuirements**:\n - This new {Created Prompt} should belong to the same domain as the {Given Prompt} but with different details.\n - The LENGTH and complexity of the {Created Prompt} should be similar to that of the {Given Prompt}.\n - The {Created Prompt} must be reasonable and must be understood and responded by humans.\n - If the {Given Prompt} includes a specific input as part of its instructions, create a new input for your {Created Prompt} when applicable.\n **Constraints that must be followed**:\n - The {Created Prompt} should be self-contained, with **all necessary information** provided, so that it can be responded to without needing to refer back to the {Given Prompt}.\n - Your response should contain **only** the {Created Prompt}, **without any** additional formatting or introductory phrases such as 'Here is the created prompt:' or 'The created prompt is:'.\n The {Given Prompt}: \n<PROMPT>\n ========\nBased on the prompt above, create your prompt:\n{Created Prompt}:\n` |

*Table 10.* Example from SimPO's preference dataset, in which the explicit reward margin and implicit reward margin are both large. We rescaled the explicit reward value to a comparable range of implicit rewards.

| | |
|---|---|
| **Prompt $x$** | Given the task definition and input, reply with output. In this task, you are given Wikipedia articles on a range of topics as passages and a question from the passage. We ask you to answer the question by classifying the answer as 0 (False)or 1 (True)\n\nPassage: Tabasco sauce -- Tabasco sauce is a brand of hot sauce made exclusively from tabasco peppers (Capsicum frutescens var. tabasco), vinegar, and salt. It is produced by McIlhenny Company of Avery Island, Louisiana.\nQuestion: is tabasco sauce and hot sauce the same\n |
| **Chosen Response $y_w$** | 0 (False)\n\nWhile Tabasco sauce *is* a type of hot sauce, the passage states that Tabasco sauce is a brand name. Therefore, not all hot sauce is Tabasco sauce. \n |
| **Rejected Response $y_l$** | 1 \n |
| **Explicit Rewards** | Chosen: $r(x, y_w) = 10.3$, rejected: $r(x, y_l) = 3.4$, explicit reward margin: $M_r(x, y_w, y_l) = 6.9$. |
| **Implicit Rewards** | Chosen: $\hat{r}_\theta^{\text{Sim}}(x, y_w) = -1.4$, rejected: $\hat{r}_\theta^{\text{Sim}}(x, y_l) = -7.7$, implicit reward margin: $M_\pi(x, y_w, y_l) = 6.3$. |

*Table 11.* Example from SimPO's preference dataset, in which the explicit reward margin and implicit reward margin are both small. We rescaled the explicit reward value to a comparable range of implicit rewards.

| | |
|---|---|
| **Prompt $x$** | let's play a game. i say a sentence, then you make a sentence that follows up my sentence then i give a continuation to yours and so on. you ready? |
| **Chosen Response $y_w$** | Sounds fun! I'm ready. Give me your first sentence. \ud83d\ude0a \n\n |
| **Rejected Response $y_l$** | I'm ready! Let's do it. \ud83d\ude0a \n\n**Give me your first sentence!** \n\n |
| **Explicit Rewards** | Chosen: $r(x, y_w) = 13.7$, rejected: $r(x, y_l) = 13.0$, explicit reward margin: $M_r(x, y_w, y_l) = 0.7$. |
| **Implicit Rewards** | Chosen: $\hat{r}_\theta^{\text{Sim}}(x, y_w) = -3.7$, rejected: $\hat{r}_\theta^{\text{Sim}}(x, y_l) = -2.9$, implicit reward margin: $M_\pi(x, y_w, y_l) = 0.8$. |

*Table 12.* Example from SimPO's preference dataset, in which the reward model's judgment is incorrect. In this example, the text provided in the prompt does not explicitly state what Zulu soldiers are called, thus the correct answer to the question *"What are Zulu soldiers called?"* would be *"No Answer"*, which corresponds to the "rejected" response by the reward model. Note that the response with a higher implicit reward is correct in this example (we use SimPO's implicit reward in implementation as explained in §3.2).

| | |
|---|---|
| **Prompt $x$** | Please answer the following question: Please answer the following question about this movie plot. If it's un-answerable, please output \"No answer\". Question: What are Zulu soldiers called? Movie plot title: Zulu Dawn Movie plot: The film is set in British South Africa, in the province of Natal, in January 1879. The first act of the film revolves around the administrators and officials of Cape Colony, notably the supremely arrogant Lord Chelmsford and the scheming Sir Henry Bartle Frere, who both wish to crush the neighbouring Zulu Empire, which is perceived as a threat to Cape Colony's emerging industrial economy. Bartle Frere issues an impossible ultimatum to the Zulu king, Cetshwayo, demanding that he dissolve the Zulu Empire. Cetshwayo refuses, providing Cape Colony with a pretext to invade Zululand. Despite objections from leading members of Cape Colony's high society and from Great Britain itself, Bartle Frere authorises Lord Chelmsford to lead a British invasion force into Zululand. The rest of the film focuses on the British invasion of Zululand and the lead-up to the Battle of Isandlwana. The invading British army, laden with an immense network of supply wagons, invades Zululand and marches in the direction of Ulundi, the Zulu capital. British forces, eager to fight a large battle in which they can unleash their cutting-edge military technology against the vast Zulu army, become increasingly frustrated as the main Zulu army refuses to attack the British, and fighting is restricted to a few small skirmishes between British and Zulu scouts. Concerned that their supply lines are becoming overstretched and that the main Zulu army is still at large, British troops begin torturing captive Zulu warriors in an effort to learn the location and tactics of the Zulu army. Halfway to Ulundi, Chelmsford halts his army at the base of Mount Isandhlwana, ignoring the advice of Boer attendants to entrench the camp and laager the supply wagons, leaving the camp dangerously exposed. During the night, Colonel Durnford and an escort of fifty mounted Basutos approach the camp. Lord Chelmsford then orders Durnford to return to his unit, bringing them to the camp immediately to reinforce Colonel Pulleine. Lt. Vereker should join Durnford as...\nA: |
| **Chosen Response $y_w$** | Impis \n |
| **Rejected Response $y_l$** | This plot description does not state what Zulu soldiers are called. No answer. \n |
| **Explicit Rewards** | Chosen: $r(x, y_w) = 11.2$, rejected: $r(x, y_l) = 5.0$, explicit reward margin: $M_r(x, y_w, y_l) = 6.2$. |
| **Implicit Rewards** | Chosen: $\hat{r}_\theta^{\text{Sim}}(x, y_w) = -8.9$, rejected: $\hat{r}_\theta^{\text{Sim}}(x, y_l) = -3.4$, implicit margin: $M_\pi(x, y_w, y_l) = 5.5$. |

