# OpenReview forum: "Larger or Smaller Reward Margins to Select Preferences for LLM Alignment?"
_ICML.cc/2025/Conference — ICML 2025 poster_

### Official Review · Reviewer_bqyD · 2025-03-09

**Overall Recommendation:** 2

**Summary:**

This paper identifies that existing metrics for selecting preference data, which rely on either explicit reward margins or implicit reward margins, often yield contradictory evaluations for the same dataset. To address this issue, the authors propose a new metric called the alignment potential ($M_{AP}$), which quantifies the gap between a model’s current implicit reward margin and a target explicit reward margin. Empirical results demonstrate that training on data selected by $M_{AP}$ consistently enhances alignment performance, outperforming existing metrics. Furthermore, the proposed metric can be integrated into self-play data generation frameworks.

## Update After Rebuttal

Thanks for the authors' effort during the rebuttal. I still have concerns regarding the computation of the implicit reward margin. As the authors replied, “The implicit margins, $M_\pi = |\hat r_\theta(x,y_w) - \hat r_\theta(x,y_l)|$, are indeed computed based on the initial model (i.e., the SFT model $\pi_\text{SFT}$).” However, based on prior work and empirical observations, an SFT model typically assigns very similar implicit rewards to different responses, indicating a lack of meaningful discrimination. Thus, I remain skeptical about this formulation. While I understand that the implicit reward may gradually become more distinguishable as training progresses, the initial SFT model itself is generally not capable of such fine-grained reward estimation. In fact, upon revisiting the cited paper [1], I found that the implicit reward in that work was not computed using the SFT model, but rather using a trained model, with the SFT model serving only as a reference. A raw SFT model, not yet trained via DPO or other preference optimization methods, lacks the necessary discriminatory ability. This remains my primary concern with the paper. I would appreciate a discussion on this issue in future revisions, particularly addressing how the results might differ if a trained model were used to compute the implicit reward instead of an untrained SFT model, especially for both the baseline and the proposed method.

[1] Not All Preference Pairs Are Created Equal: A Recipe for Annotation-Efficient Iterative Preference Learning

**Claims And Evidence:**

1. The authors claim that "while existing metrics primarily assess data quality based on either explicit or implicit reward margins, they often provide contradictory evaluations for the same data" as the main motivation of this paper. However, this claim is only illustrated through two specific cases in Figure 1(a). There is a lack of quantitative analysis to determine how frequently such contradictions occur in commonly used preference datasets. Conducting a quantitative study on the proportion of "contradictory" data points in widely used datasets would strengthen the motivation of this work.

2. It is unclear whether the implicit reward margins used in this work are computed from a policy that has already been optimized on the preference dataset, or if they are all derived directly from an SFT model as mentioned on Page 5. If the latter is the case, it raises concerns, as an SFT model is expected to be relatively neutral in its preferences for all the preference data. This leads to the question of how the distribution of implicit reward margins differs before and after optimization? Additionally, in line with the first concern, it would be useful to quantify the proportion of data points where implicit reward margins contradict explicit reward margins. The choice of implicit reward margins is crucial to the paper’s claims—why should smaller implicit reward margins be preferable?

**Essential References Not Discussed:**

NA

**Experimental Designs Or Analyses:**

1. In Table 1, the primary experiments only compare $M_{AP}$ against baselines that use explicit reward margins $M_r$ for data selection. However, there is no baseline that selects data based on implicit reward margins $M_\pi$.

2. The experiments mention the hyperparameter $\alpha$ for the proposed $M_{AP}$, but do not report the specific values used across different experiments. Providing this information would improve the reproducibility of the results.

**Methods And Evaluation Criteria:**

Is the proposed method limited to SimPO, or can it generalize to other approaches? For instance, in standard DPO, the implicit reward margins computed from an SFT model are always zero. How would this affect the applicability of the proposed $M_{AP}$ metrics in such settings?

**Other Comments Or Suggestions:**

NA

**Other Strengths And Weaknesses:**

Overall, this paper is well-organized and easy to read. However, as noted in the Claims and Evidence and Experimental Designs or Analyses sections, there are still some concerns regarding the lack of quantitative validation for key claims and missing baselines.

**Questions For Authors:**

Please see above. I am very happy to discuss with the authors during the rebuttal process and look forward to having the above concerns and questions addressed.

**Relation To Broader Scientific Literature:**

Can you provide a theoretical or intuitive explanation of why a larger explicit reward margin is always better, while a smaller implicit reward margin is preferable? A clear justification for this assumption would strengthen the correlation to previous works and strengthen the motivation of this paper.

**Theoretical Claims:**

Most of the theoretical results appear to be correct upon review. However, the detailed proofs provided in the appendix were not fully verified.

---

> ### Author Rebuttal · Authors · 2025-03-31
>
> ## Contradictions
> We have conducted an analysis to measure the contradictions between explicit and implicit margin metrics ($M_r$ and $M_\pi$) by comparing the Jaccard similarity of subsets selected by each metric:
> - **Subset selection**: Using $M_r$ and $M_\pi$, we select the top-rated k% subsets from existing datasets, denoted as $D_r$ and $D_\pi$.
> - **Jaccard similarity**: We calculate the similarity between the selected subsets using:
> $$
> J(D_1,D_2) = \frac{|D_1 \cap D_2|}{|D_1 \cup D_2|}.
> $$
> - **Comparison**: We compare the resultant similarity $J(D_r, D_\pi)$ with two uniformly selected subsets $J(D_{u1},D_{u2})$. A lower similarity $J(D_r, D_\pi) < J(D_{u1},D_{u2})$ would indicate contradictions between $M_r$ and $M_\pi$.
>
> We measure the similarities on all three datasets in our paper:
> |Jaccard (%)|Top-5%|Top-10%|Top-20%|
> |-|:-:|:-:|:-:|
> |Gemma|1.28|3.16|7.67|
> |Llama v1|2.12|4.82|10.59|
> |Llama v2|1.42|3.46|8.25|
> |_Uniform_|2.62|5.44|11.08|
>
> As shown in the table, the similarities $J(D_r, D_\pi)$ are **consistently smaller than those of uniformly selected subsets** across varying top-k ratios and datasets. This observation validates the claim that explicit and implicit margin metrics can **indeed provide contradictory evaluations for preference selection**.
>
> ## Smaller Implicit Margins
> Explanation of implicit reward margins:
> - **Computation**: The implicit margins, $M_\pi = |\hat r_\theta(x,y_w) - \hat r_\theta(x,y_l)|$, are indeed computed based on the initial model (i.e., the SFT model $\pi_\text{SFT}$).
> - **Motivation**: **The implicit margin shows the LLM's ability to discern preferences**. A small implicit margin indicates the model initially lacks the ability to distinguish such preference, making the data suitable for alignment training.
>
> **Empirical evidence**: We measure the distributions of implicit margins before and after the alignment training on the Llama v2 dataset.
> The results, as shown in Figure I in this [anonymous link](https://anonymous.4open.science/r/tables-11915/bqyD.md), demonstrate a notable increase in $M_r$ after alignment training.
> Specifically, the average $M_r$ rises from `0.072` to `0.742`, indicating **improved preference recognition ability** of the model.
>
> Given these insights, data with relatively smaller initial implicit margins generally indicates that **the LLM cannot tell the preference, thereby requiring further training** to align on such data.
>
> ## Method Implementation
> Our SimPO-based implementation of the implicit reward is because:
> 1. The implicit reward $\hat r_\theta^{Sim}(x,y) = \beta\log\frac{\pi_\theta(y|x)}{|y|}$, has been demonstrated to be effective in SimPO's paper.
> 2. The formulation does not require additional conditions (e.g. reference model), allowing for its direct application in different optimization methods, e.g., DPO, IPO.
>
> Throughout our paper, **the implicit reward in our $M_{AP}$ metric is consistently calculated using the SimPO formulation** $\hat r_\theta^{Sim}(x,y),$  regardless of whether the training method is SimPO or DPO.
> And the empirical results also validate its effectiveness across different training settings.
>
> ## More Baselines
> In response to your suggestions, we conduct additional experiments to include uniform and $M_\pi$ baselines under Table 1's setting, and here are the results of Llama models on Alpaca Eval:
>
> |Llama+DPO|Uniform|$M_\pi$|$M_{AP}$|
> |-|:-:|:-:|:-:|
> |Alpaca LC|40.86|42.00|**42.82**|
> |Alpaca WR|42.66|44.71|**45.00**|
>
> |Llama+SimPO|Uniform|$M_\pi$|$M_{AP}$|
> |-|:-:|:-:|:-:|
> |Alpaca LC|40.69|42.39|**43.76**|
> |Alpaca WR|38.77|41.62|**42.00**|
>
> As shown in the tables, our $M_{AP}$ metric consistently outperforms existing baselines for both optimization methods.
>
> ## Hyperparameters
> The value of hyperparameter $\alpha$ can be found in Line 991 and Line 1008 of Appendix D.1, where we generally set the value within {0.5, 1, 2.5, 5.0}.
>
> ## Explain Margins & Motivation
>
> We would like to clarify that our paper is not centered on the notion that "large explicit reward margin" and "small implicit reward margin" **individually** imply high-quality data.
> Instead, one key motivation for our work is that **relying solely on either of these margins is insufficient** for assessing data quality.
> As demonstrated in Figure 1, the two instances with high explicit reward margins or small implicit margins are evaluated as *low-quality* data by our metric (Line 71-74).
>
> The core idea of our metric is to quantify the data quality by measuring **the discrepancy between the current model's preference and the aligned optimum** on the data.
> The alignment optimum is captured through explicit reward margins (Equation 8), and the model's preference is indicated by implicit reward margins.
> So the proposed metric $M_{AP} = |r(x,y_w) - r(x,y_l)|-|\hat r_\theta(x,y_w) - \hat r_\theta(x,y_l)|$ quantifies the **gap from the current implicit margin to the target explicit margin**, thereby indicating the potential for alignment on the given data.

---

### Official Review · Reviewer_ZHMc · 2025-03-13

**Overall Recommendation:** 2

**Summary:**

This paper proposes the new "alignment potential metric" to evaluate the quality of (and select) data for offline preference optimization. This metric quantifies the gap between the model's implicit reward margin and the target explicit reward margin, and thus aims to estimate the model's potential to align on the preference data by measuring how much the model can improve its preference discrimination. Using standard DPO and SimPO preference learning setups, the paper compares this preference data selection metric against existing baseline metrics (e.g., using only the model's implicit reward margin, and only the explicit reward margin), and shows that this alignment potential metric consistently outperforms. The paper also provides a theoretical justification for why training converges faster when selecting data using the alignment potential metric, versus random data selection.

**Claims And Evidence:**

A major claim is that "large explicit reward margin" and "small implicit reward margin" both imply "high quality" datapoints for preference optimization. These claims are substantiated by citations in the introduction, but since they are both very general claims and are central to the motivation for the paper's new "alignment" potential metric", it would be more convincing if the results and/or theoretic basis supporting these claims were at least summarized in the paper, in the context of the paper's own setup. Especially the claim that data for preference optimization should be selected via the "small implicit reward margin" metric is not obvious; e.g., perhaps a curriculum learning setup where the allowed implicit reward margin increases over the course of training would outperform. It would be informative to include an ablation to show that selecting datapoints for preference optimization using "small explicit reward margin" and "large implicit reward margin" metrics does not work well.

Another related and central claim in the paper is that a larger gap between the target explicit reward margin and current implicit reward margin indicates a greater potential for improvement. But the paper does not address the potential limitation that if the gap is too large, learning the correct alignment may be out of reach of the model (label exposure bias problem).

**Essential References Not Discussed:**

While the paper cites related work in preference optimization dataset selection, it does not cite other works which investigate metric-guided data selection for training LLMs more generally. For example, [this paper](https://arxiv.org/pdf/2311.05350) (among others) shows that using QE metrics for data filtering improves the quality of machine translation models. The proposed M_{AP} method also uses metric-guided filtering, just in the (paired) preference data setting.

**Experimental Designs Or Analyses:**

- The paper compares using M_{AP} against using only the model's implicit reward margin, and only the explicit reward margin, as data selection metrics. The paper does not address whether the relative magnitude of the implicit versus explicit margins is important, though, or whether it is primarily the ranking (dis)agreement that matters. Another insightful baseline would be to compare using M_{AP} for data selection against using a ranking (rather than margin) metric, which selects all examples for which the explicit and implicit rewards disagree on the ranking of the preferred versus dispreferred response. (Within this subset, examples could be randomly chosen to match the expected dataset size.)
- How does data selection with M_{AP} compare to weighting the loss by the implicit margin? And can train-time loss weighting provide incremental improvements on top of offline data selection with M_{AP}?
- The proposed M_{AP} metric selects a fixed set of examples offline before preference learning (or during multiple iterations of training, as in Section 5). However, the paper does not explore a curriculum learning setup. How does performance from selecting a fixed set of examples with a large M_{AP} margin compare to (smoothly) varying the reward margin of the examples over the course of training?
- How long were the DPO/SimPO models trained for, and how was the checkpoint to use for evaluation chosen?

**Methods And Evaluation Criteria:**

- The M_{AP} metric crucially depends on (the quality of) an external reward model, and measures the extent to which the model's preference margin matches the ground truth margin as given by this external reward. The effect of the reward model used for data selection is not explored in the main text.
  - To what extent does the quality of the M_{AP} metric depend on the quality of the explicit reward model? (This model is being considered as a proxy for the gold rewards.)
  - An assumption in the formulation of the M_{AP} metrics is that all reward margins of equal magnitude imply an equal quality gap. But in practice, the mapping between quality and reward model score is often non-linear, and a large reward margin at the upper versus lower end of the quality spectrum may imply different actual differences in quality.
  - Can the same LLM generating the responses be used as a prompted LLM-as-a-Judge reward model to get the "explicit rewards"? If not, what are the required characteristics of the external reward model for this method to yield gains?
- The "Reward noise regularization" subsection (Section 3.1) is poorly motivated. The regularization simply consists in taking the absolute value of the explicit and of the implicit margins. In the extreme (for a certain explicit reward model), this metric could select examples for which the model always agrees with the external reward about the ranking of the preferred and dispreferred responses, which is intuitively the opposite of what M_{AP} purports to do. So the effectiveness of this regularization is likely highly dependent on the characteristics of the external reward being used.
- The benchmarks and evaluation metrics used were reasonable, and are standard in the preference optimization literature.

**Other Comments Or Suggestions:**

- The M_{AP} equation is repeated multiple times throughout the paper. Is this necessary, or can equation (11) just be referenced?
- Heavily overlapping results between Figure 1b and Figures 4/5
- The Discussion section contains nothing more than the Conclusion + Limitations and future work.

**Other Strengths And Weaknesses:**

Strengths:
- The results look strong. The proposed M_{AP} method for preference data selection consistently outperforms other metrics, across several experiments in which data is selected both from fixed and iteratively evolving datasets.
- The paper shows that by selecting data with M_{AP}, training converges faster than with baseline data selection methods. Moreover, the performance gains are consistent across dataset sizes.

Weaknesses:
- The proposed M_{AP} metric is extremely simple and not very novel (very small incremental contribution on top of existing work).
- There is no significance testing to accompany the experimental results (and, for example, the results using M_{pi} versus M_{AP} in Figure 5 look very close).
- The Introduction is repetitive and introduces concepts/intuitions which are redundant with those presented in the Methods section (Section 3).
- The entire Related Work section is in the Appendix.

**Questions For Authors:**

No additional questions besides those posed in previous sections (especially see the "Methods And Evaluation Criteria" and "Experimental Designs Or Analyses" sections).

**Relation To Broader Scientific Literature:**

This paper contributes to the body of literature on data selection for post-training/alignment. There has been a lot of recent interest and work in this area, not only for selecting datasets for preference optimization (e.g., DPO), but also for selecting SFT and RL datasets.

**Theoretical Claims:**

Theoretical results (Theorem 3.1, proof in Appendix) show that training converges faster when using M_{AP} for data selection, vs random sampling. This result is novel and insightful.

---

> ### Author Rebuttal · Authors · 2025-03-31
>
> Due to space limits, we put all tables in this anonymous link: https://anonymous.4open.science/r/tables-11915/ZHMc.md.
>
> ## Large/Small Margins
> To reiterate, the primary claim of our paper is **not** that "large explicit reward margin" and "small implicit reward margin" **individually** imply high-quality data.
> Instead, the key motivation for our work is that **relying solely on either of these margins is insufficient** for assessing data quality.
>
> As demonstrated in Figure 1, we evaluate instances with high explicit reward margins or small implicit margins as low-quality data (Line 71-74).
> The main idea of our proposed metric is to consider the **gap between the model's current implicit margin and the target explicit margin** to measure the potential for alignment learning.
>
> However, if assessing data quality using a single margin is *required*, we acknowledge that a higher explicit margin or smaller implicit margin can be beneficial. This notion is supported by recent works (Line 40) and our experiments (Figures 4, 5, and 8).
> In response to your suggestions, we've conducted additional experiments by following the setup of Figure 4, but **reversely selecting data** via these metrics.
> As shown in **Table I of the linked page**, reversely selecting training data indeed results in poorer performance, thus verifying these metrics' efficacy from an opposite standpoint.
>
> ## Too Large Gaps
> While a larger gap might require more intensive alignment training, the direction of the update remains correct.
> Moreover, the common practice in preference dataset construction for alignment training, such as SimPO, involves **sampling responses $y_w$ and $y_l$ using the model that is being trained**.
> This practice ensures that both the chosen and rejected responses are **within the model's generation capabilities**, thus mitigating the label exposure bias problem.
>
> ## Different Reward Models
> In our study, we have indeed considered the impact of different reward models by incorporating two RMs: **PairRM (0.4B) and ArmoRM (8B)**, for preference annotation and metrics evaluation.
> Figures 4 and 8a in the paper illustrate the results based on datasets annotated using ArmoRM and PairRM, respectively.
> As shown in the figures, **our proposed $M_{AP}$ metric outperforms existing baselines under both RMs**, attesting to the effectiveness of $M_{AP}$ metric under different reward models.
>
> ## Reward Noise & Ranking
> **The ranking disagreement might not be the key determining the data quality**, due to the reward noise issue.
> As suggested, we select 40% data subsets in which the ranking of explicit and implicit rewards are opposite.
> We compare this strategy with existing metrics on both Llama datasets as described above.
>
> As shown in **Table II of the linked page**, although selecting data with **contradictory ranking** between explicit and implicit rewards can improve performance to some extent, it **cannot outperform any of the existing baselines**.
>
> Such an observation is consistent with our motivation of "Reward noise regularization".
> As discussed in Line 212-215, given the LLM's ability to discern certain preferences, **the opposite ranking** between explicit and implicit rewards—suggesting contradictory preference judgments between the RM and the LLM—may indicate **a higher risk of noisy reward model annotations**, and thus resulting inferior model performance.
>
> ## Other Directions
> We sincerely appreciate your suggestions for exploring the *non-linear mapping, LLM-as-Judge, weighted loss and curriculum learning* settings.
> While we believe they hold promise for future research, the limited timeframe for the current study does not permit us to explore all these directions, so we plan to investigate them in future work.
>
> ## Novelty
> While existing methods focus solely on either explicit or implicit margins and often lack in-depth explanations, our work introduces a key innovation by measuring the data quality with **the gap between the model's aligned optimum and the current model's preferences**.
> This gap serves as a **theoretical basis** for evaluating data quality in alignment training, distinguishing our approach from prior work.
>
> Although the final form of $M_{AP}$ incorporates the two existing metrics ($M_r$ and $M_\pi$), **its core idea of gap quantification and the logical derivation of our approach are novel aspects not explored in existing metrics**., which contributes significantly to the advancement and originality of this work.
>
> ## Training Details & Significance Testing
> We report the training time, standard deviations and confidence intervals of current evaluations in **Table III & IV of the linked page**.
>
> ## References & Presentation
> Thanks for your suggestions, we will make the following revisions:
> 1. Include relevant references of data selection in general LLM tasks
> 2. Merge repetitive equations and figures
> 3. Revise sections by splitting the Discussion section into separate Conclusion and Limitations sections

---

### Official Review · Reviewer_66Hn · 2025-03-23

**Overall Recommendation:** 4

**Summary:**

This paper examines how reward margins influence preference data selection in LLM alignment. It introduces a novel metric, Alignment Potential (AP), which integrates both explicit reward margins (provided by the reward model) and implicit reward margins (derived from the policy and reference models). Furthermore, it extends AP to self-play data generation. Extensive experiments demonstrate that AP outperforms leading alignment baselines across various settings.

**Claims And Evidence:**

The claims are well-supported by both theoretical analysis and experimental results.

**Essential References Not Discussed:**

To my knowledge, this paper has included sufficient references.

**Experimental Designs Or Analyses:**

The authors first conduct preliminary experiments to assess different metrics (explicit reward margin, implicit reward margin, and AP) in the context of LLM alignment, supported by theoretical analysis. The experiments are then extended to a self-play data generation framework, with clear comparisons against baseline methods (DPO, SimPO with different self-play strategies) across various model architectures (Llama3-8B, Gemma2-9B) and dataset sizes (10k–60k). The experimental design is thorough and convincingly demonstrates AP’s effectiveness.

**Methods And Evaluation Criteria:**

The proposed method is well-aligned with the LLM alignment problem, focusing on contrast strategies in preference data selection—specifically, whether larger or smaller reward margins should be used. AP is a theoretically and empirically grounded metric for assessing preference data quality.

To evaluate AP’s effectiveness, the paper employs well-established preference optimization benchmarks (ApacaEval2, Arena-Hard) and diverse model architectures (Llama3-8B, Gemma2-9B).

**Other Comments Or Suggestions:**

See weaknesses.

**Other Strengths And Weaknesses:**

Strengths

-	The paper is well-written, with clear motivation and an intuitive structure.
-	It establishes connections between existing data selection strategies (explicit and implicit reward margins).
-	Preliminary experiments, paired with theoretical analysis, support AP’s effectiveness.
-	Extensive experiments consistently show AP’s superior performance across LLM architectures, benchmarks, and dataset sizes. Notably, models trained on only AP-selected data (e.g., 30k samples) outperform those trained on full-sized datasets.

Weaknesses

-	AP is tested only with DPO and SimPO. While the experiments are thorough and convincing, evaluating AP with additional alignment approaches would strengthen the results.

**Questions For Authors:**

See weaknesses.

**Relation To Broader Scientific Literature:**

1.	This paper builds on research in LLM alignment, particularly in direct preference optimization (DPO) and self-play preference generation.
2.	It explores a critical problem in LLM alignment—how contrast strategies (explicit and implicit reward margins) in preference data selection impact alignment. AP is proposed as a novel strategy that integrates these reward margins to provide a more effective metric.
3.	Beyond empirical validation, the paper provides theoretical analysis to justify AP’s ability to identify high-quality alignment data.

**Theoretical Claims:**

The proofs, including those in the appendix, have been reviewed, and no issues were found.

---

> ### Author Rebuttal · Authors · 2025-03-31
>
> ## Additional Alignment Approaches
>
> Thanks for your suggestion!
> Aside from the DPO and SimPO methods, we conduct additional experiments using the IPO method for alignment training.
>
> Like the setting in Section 3.2, we utilize different metrics to select top-40% subsets from existing preference datasets for subsequent IPO training.
> We utilize the two Llama-based datasets for experiment, benchmark the trained models on Alpaca Eval 2.0, and report both Length-Controlled (LC) win rates and raw Win Rates (WR):
>
> On the default Llama dataset of SimPO (based on the PairRM reward model):
>
> | **Llama+IPO** | Uniform | $M_r$ | $M_\pi$ | $M_{AP}$ (ours) |
> |---|:---:|:---:|:---:|:---:|
> | Alpaca LC | 30.49 | 34.15 | 35.04 | **35.44** |
> | Alpaca WR | 34.15 | 34.08 | 35.40 | **36.08** |
>
> On the Llama v2 dataset of SimPO (based on the ArmoRM reward model):
>
> | **Llama-v2+IPO** | Uniform | $M_r$ | $M_\pi$ | $M_{AP}$ (ours) |
> |---|:---:|:---:|:---:|:---:|
> | Alpaca LC | 33.17 | 33.92 | 38.19 | **40.35** |
> | Alpaca WR | 30.63 | 30.17 | 35.73 | **38.14** |
>
> As shown in the table, our proposed $M_{AP}$ metrics still **outperform existing baselines under this new alignment method**, further certifying its effectiveness.
>
> **Reference**
> - IPO: Azar, Mohammad Gheshlaghi et al. “A General Theoretical Paradigm to Understand Learning from Human Preferences.” ICML 2024.

---

### Official Review · Reviewer_JxER · 2025-03-24

**Overall Recommendation:** 3

**Summary:**

This paper investigates the techniques of preference data selection, particularly between the explicit reward margin by reward models and the implicit reward margin by the SFT models. This paper proposes a new preference pair quality metric (MAP) based on the discussion between the above two reward margins, enforcing the high explicit reward margin and the low implicit reward margin simultaneously. The experiments show that, using MAP as the preference data selector, alignment quality is improved compared with other selection methods.

**Claims And Evidence:**

Yes.

**Essential References Not Discussed:**

N/A

**Experimental Designs Or Analyses:**

1. The comparison between different preference data selection methods.

2. GPT-4 agreement on the selected preference pairs.

3. The DPO and SimPO alignment results in the self-play setting.

**Methods And Evaluation Criteria:**

The proposed MAP data selector is clear and easy to follow. The evaluation designs are convincing.

**Other Comments Or Suggestions:**

In L216, what is the M_+ method? It is not clearly stated before this line.

**Other Strengths And Weaknesses:**

1. Although the scope of this paper is quite narrow, this paper still provides an in-depth discussion of the reward margins, which I think is helpful for readers to understand the literature.

2. I am surprised by the simplicity of the proposed method. I think that is the reason why this paper focuses on too many technical details and is a little bit hard to read.

**Questions For Authors:**

N/A

**Relation To Broader Scientific Literature:**

This paper provides an easy yet effective preference data selection method, which is essential for constructing good preference datasets.

**Theoretical Claims:**

I did not check the correctness of DPO convergence, since I am not an expert in this field.

---

> ### Author Rebuttal · Authors · 2025-03-31
>
> ## The $M_+$ Metric
>
> The $M_+$ metric is a modified version of the proposed $M_1$ metric designed to adapt to the **unidirectional update nature of existing preference optimization methods** (e.g. DPO, SimPO).
>
> As described in section 3.1, we first quantify the discrepancy between the current model and the aligned optimum on the preference data $(x,y_w,y_l)$ via the $M_1$ metric:
> $$M_1(x,y_w,y_l;\theta,r^*) = |(r^*(x,y_w) - r^*(x,y_l)) - (\hat r_\theta(x,y_w) - \hat r_\theta(x,y_l))|.$$
> To ensure practicality, we then introduce two key adaptations to the $M_1$ metric,  and the first one is termed **unidirectional calibration** (Lines 182-202). The rationale behind this adaptation is explained as follows:
>
> In the optimization process of DPO, the loss function $-\log\sigma(\hat r_\theta(x,y_w) - \hat r_\theta(x,y_l))$ tends to **unidirectionally increase the implicit margin term** $\hat r_\theta(x,y_w) - \hat r_\theta(x,y_l)$ during optimization.
> So we would like to select data with a low implicit margin term requiring to be increased through training.
>
> However, the original $M_1$ metric measures a **bidirectional gap** between $\hat r_\theta(x,y_w) - \hat r_\theta(x,y_l)$ and $r^*(x,y_w) - r^*(x,y_l)$, which can lead to selection of data with either a very high margin or a very low margin.
> While data with a low margin term, $\hat r_\theta(x,y_w) - \hat r_\theta(x,y_l)$, is desirable,
> data with a high margin term will be further increased by the DPO training process, which is counterproductive.
>
> To address this, we opted to remove the absolute value from $M_1$, implementing what we refer to as unidirectional calibration, and formulated a revised metric, $M_+$:
> $$M_+(x,y_w,y_l;\theta,r^*) = (r^*(x,y_w) - r^*(x,y_l)) - (\hat r_\theta(x,y_w) - \hat r_\theta(x,y_l)).$$
>
> This new metric effectively selects data with a minimal implicit margin term $\hat r_\theta(x,y_w) - \hat r_\theta(x,y_l)$ to increase, **aligning with the unidirectional nature of preference optimization methods**.
>
> We hope this elucidation clarifies the intent and application of the $M_+$ metric within our study.
>
> ## Readability
>
> Thank you for highlighting this issue.
> We will strive to enhance the readability in our revised version.
> Specifically, we plan to: (1) merge the two $M_{AP}$ equations (Eq. 11 and Line 236) to reduce redundancy and enhance clarity; (2) include a concise overview of our metric formulation at the beginning of Section 3 to guide readers through our methodology with greater ease.
> These changes aim to streamline the presentation and make the content more coherent to readers.

---

> > ### Comment · Reviewer_JxER · 2025-04-07
> >
> > Thanks for the response! My concerns have been addressed.

---

### Official Review · Reviewer_gdV3 · 2025-03-25

**Overall Recommendation:** 4

**Summary:**

The submission addresses the challenge of selecting high-quality preference data for aligning large language models with human values. It introduces the metric of alignment potential. This metric quantifies the gap between a model’s current implicit reward margin and the target explicit reward margin to identify preference pairs with high potential for improving alignment. By integrating this metric into both offline preference learning and self-play data generation, the methodology effectively selects data that accelerates convergence and enhances overall performance, achieving superior alignment and robust performance gains compared to traditional data quality metrics.

**Claims And Evidence:**

The primary contribution of the submission is the introduction of the Alignment Potential (MAP) metric as a method to select preferences effectively for model alignment. The authors present a clear and logical derivation of this metric, progressing from the initial metric formulation ($M_1$) through subsequent refinements ($M_+$) until the final $M_{AP}$ formulation. Furthermore, the paper provides empirical evidence demonstrating the superiority of $M_{AP}$ over baseline methods, confirming its practical benefits.

**Essential References Not Discussed:**

Not applied

**Experimental Designs Or Analyses:**

I reviewed the experimental design and analyses presented in the paper. The methodology is sound, with appropriate evaluation protocols across the relevant benchmarks. The authors provide clear comparisons between their proposed approach and existing baselines in both offline and online settings.

**Methods And Evaluation Criteria:**

The proposed $M_{AP}$ metric represents a novel approach to me for selecting high-quality data for LLM alignment. This derivation shows how combining the explicit reward margin with the implicit reward margin results in a unified metric that quantifies the model’s potential for alignment improvement. Furthermore, the empirical evaluation employs benchmark datasets and evaluation metrics that are common practice in the field.

**Other Comments Or Suggestions:**

I strongly recommend that the authors enhance Figure 1 with more insightful examples to better illustrate the potential impact of this work. The current examples fall short in conveying the broader implications, making it challenging to fully appreciate the significance of the proposed approach. To be more specific, Example I demonstrates a case with a clear ground-truth label that can be easily handled by other methods. By contrast, Example II features two very similar responses, rendering it difficult to discern which one is superior—even with human evaluation.

**Other Strengths And Weaknesses:**

The clarity of the writing in this submission makes it an engaging and enjoyable read. The logical progression from $M_1$ to $M_+$ and ultimately to $M_{AP}$ is smooth. Additionally, the authors provide an example for calculating the various metrics.

**Questions For Authors:**

Could you please explain in more detail the potential impact of addressing the issues presented in the two examples in Figure 1?

**Relation To Broader Scientific Literature:**

From my understanding, this paper makes two key contributions in broader scientific literature:
- Advances in Data Selection for Preference Optimization: The $M_{AP}$ metric refines data selection by effectively distinguishing high-quality from lower-quality preference data, which streamlines the alignment process.
- Integration of Self-Play Mechanisms: Incorporating self-play supports scalable model refinement through self-generated content, beyond static data selection. The metric enables the effective identification of high-quality preference pairs within self-generated content, supporting continuous model refinement in dynamic training scenarios.

**Theoretical Claims:**

I have carefully reviewed the proofs presented in the main text as well as those in Section C.1 of the Appendix and have found no errors.

---

> ### Author Rebuttal · Authors · 2025-03-31
>
> ## Suggestions & Questions Regarding Figure 1
>
> Thanks for your valuable suggestion!
>
> In Figure 1, we present two examples to compare the existing metrics:
> - **Explicit reward margin**: $M_r = |r(x,y_w) - r(x,y_l)|$, which quantifies **how much $y_w$ is more preferable than $y_l$**.
> - **Implicit reward margin**: $M_\pi = |\hat r_\theta(x,y_w) - \hat r_\theta(x,y_l)|$.
> With $\hat r_\theta(x,y)$ indicating the current model $\pi_\theta$'s preference evaluation on $y|x$, this measures **how well the model discerns the preference between $y_w$ and $y_l$**.
>
> While existing works select data with *larger explicit reward margins* or *smaller implicit reward margins* for training,
> we illustrate how these metrics can yield contradictory evaluations for the same data in Figure 1—demonstrating inaccuracies since at least one evaluation must be incorrect.
> 1. In **Example I**, the chosen response is clearly correct and more preferable, resulting in a substantial explicit reward margin (indicated by the blue bar), and thus deemed "high-quality" by $M_r$.
> However, the large implicit reward margin (shown by the orange bar) suggests that the current model $\pi_\theta$ can already distinguish the preferences between $y_w$ and $y_l$, thus being  "low-quality" by $M_\pi$.
> Consequently, despite what $M_r$ indicates,
> this data cannot further improve preference learning as the model is already well-aligned on this sample—**a scenario where relying on $M_r$ is misleading**.
> 2. **Example II** presents two nearly identical responses $y_w$ and $y_l$, resulting in a very small explicit reward margin, and thus being "low-quality" by $M_r$.
> Similarly, the model cannot tell the preference between such similar responses and produces a small implicit reward margin, which will be evaluated as "high-quality" by $M_\pi$.
> Given the negligible preference between the responses, this data should be considered as low-quality—**highlighting how reliance on $M_\pi$ can also be fallacious**.
>
> These examples underscore the shortcomings of relying solely on a single margin, and motivate us to derive a more propoerly designed metric:
> - Since $M_r$ meaures the difference between $y_w$ and $y_l$, it serves as an **alignment target** indicating **how the preference should be** on the specific data.
> - In contrast, $M_\pi$ corresponds to the model's **current state**, indicating **how the preference of the current model is** on this data.
> - As shown in the two examples, evidenced by the examples, data quality cannot be determined merely by a large target value ($M_r$) or a small current value ($M_\pi$); rather, **it is the gap between the current and target preferences that holds significance**.
> - From this insight, our proposed $M_{AP}$ metric evaluates preference data quality by quantifying the gap from the current implicit margin to the target reward margin, thereby measuring the potential for further alignment training.
>
> We acknowledge that the connection between the contradictions presented in Figure 1 and the core concept of the proposed $M_{AP}$ metric is not sufficiently clear in the current figure.
> To address this, we will enhance Figure 1 by including more informative text within the figure or title to explicitly highlight that existing metrics fail on these two examples because they solely focus on either the current model or the target value, neglecting the crucial gap between them.
> Thank you once again for your insightful suggestion!

---

> > ### Comment · Reviewer_gdV3 · 2025-04-06
> >
> > Thank the authors for addressing my comments and questions, and most of my concerns have been resolved.

---

### Official Review · Reviewer_HZK1 · 2025-03-25

**Overall Recommendation:** 3

**Summary:**

- This paper proposes a new way to select good preference data jointly using two different reward signals which are captured through the external reward model and training model’s implicit DPO reward, respectively. Start from the mathematical derivation of DPO, the authors suggest the revised score function to alleviate the problem from the original score. Through two different practical scenarios of preference learning (offline and online), the proposed method is demonstrated compared to the existing baselines.

**Claims And Evidence:**

Yes.

**Essential References Not Discussed:**

Current references are sufficient, but it would be nice if some works are additionally referred and discussed. Those works are mentioned in the below comments.

**Experimental Designs Or Analyses:**

Yes.

**Methods And Evaluation Criteria:**

Yes.

**Other Comments Or Suggestions:**

Please address the above concerns.

**Other Strengths And Weaknesses:**

### Pros

1. **Clarity**. Overall, the writing is clear and easy to follow. In addition, the organization of the main draft is well-established.
2. **Well-motivated problem and intuitive approach.** The selection of good preference data is an interesting problem, and the proposed approach seems to be intuitive and effective.

### Cons

- **More ablation study**. While the authors provide some ablation studies to demonstrate the effectiveness of the proposed ideas, more experiments are required for a rigorous ablation.
    - In Figure 3, the proposed score (Eq. 11) is only compared with the score in Eq. 10. However, as there are two modification (taking absolute value in external reward gap & taking absolute value in implicit reward gap), it is unclear whether they are really necessary. It would be nice if the authors can provide two additional results for each modification (e.g.,  absolute external reward gap like Eq. 11 & implicit reward gap like Eq. 10 and vice versa).
    - In section 3.2, all the experiments are conducted across different selection methods under a fixed selection ratio (top-k%, k=40). For more extensive demonstration of the proposed method, it would be nice if the authors provide the additional results by (1) selecting bottom-40% samples and (2) varying k values such as 10% and 70%.
- **More baselines for iterative preference learning setup**. In figure 6, the authors conduct the experiments about the iterative preference learning setup. As many prior works for online DPO have been proposed for this problem [1,2,3], it would be nice if the authors can demonstrate the effectiveness of the proposed method compared to them. Also, the baselines in Figure 5 (uniform, $M_r$, $M_{\phi}$ are directly applicable for this experiment, too.

[1] Xiong et al., Iterative Preference Learning from Human Feedback: Bridging Theory and Practice for RLHF under KL-constraint., ICML 2024
[2] Kim et al., Spread Preference Annotation: Direct Preference Judgment for Efficient LLM Alignment., ICLR 2025
[3] Chen et al., Bootstrapping Language Models with DPO Implicit Rewards., ICLR 2025

**Questions For Authors:**

Please address the above concerns.

**Relation To Broader Scientific Literature:**

The key contributions are about new ideas and results.

**Theoretical Claims:**

Yes.

---

> ### Author Rebuttal · Authors · 2025-03-31
>
> ## Ablation I: absolute values
>
> Although Eq.11 incorporates two additional absolute values compared with Eq.10: $|r(x,y_w) - r(x,y_l)|$ and $|\hat r_\theta(x,y_w) - \hat r_\theta(x,y_l)|$, only the latter absolute values (on implicit rewards) actually changes the metric's value.
> The reason is that, when constructing the preference dataset $\{(x,y_w,y_l)\}$, two responses $y_1,y_2$ are annotated by the reward model $r$ to determine the winning/losing responses: $y_w,y_l \in \{y_1,y_2\}$, such that $r(x,y_w) > r(x,y_l)$  *(line 182-185)*.
> Therefore, by definitation we have $r(x,y_w) - r(x,y_l) \ge 0$ and $|r(x,y_w)-r(x,y_l)| = r(x,y_w) - r(x,y_l)$.
> Given that **the absolute value applied to the external reward gap does not alter the outcome**, there will be no need to conduct additional ablations.
>
> We will explicitly clarify this point in our revised manuscript to prevent potential confusion.
> Thank you for pointing out this problem!
>
> ## Ablation II: Varying-k & Bottom-40%
>
> Thanks for your insightful suggestions!
> We have conducted additional experiments to address your feedback on varying selection ratios and the impact of selecting bottom-40% samples.
>
> For varying k values, we expanded our experiments by selecting different top-k on the default SimPO's Llama v1 dataset using various metrics, and evaluated the resulting models using Alpaca Eval 2.0.
> We reported both Length-Controlled (LC) win rates, which are more preferable to reduce length bias, and raw Win Rates (WR):
>
> | **Select 20%** | Uniform | $M_r$ | $M_\pi$ | $M_{AP}$ (ours) |
> |---|:---:|:---:|:---:|:---:|
> | Alpaca LC | 26.02 | 28.95 | 28.46 | **30.66** |
> | Alpaca WR | 28.06 | 30.24 | **31.01** | 30.62 |
>
> | **Select 40%** | Uniform | $M_r$ | $M_\pi$ | $M_{AP}$ (ours) |
> |---|:---:|:---:|:---:|:---:|
> | Alpaca LC | 30.72 | 36.58 | 33.53 | **37.07** |
> | Alpaca WR | 32.64 | 36.12 | 34.75 | **36.58** |
>
> | **Select 60%** | Uniform | $M_r$ | $M_\pi$ | $M_{AP}$ (ours) |
> |---|:---:|:---:|:---:|:---:|
> | Alpaca LC | 33.94 | 40.77 | 38.29 | **42.86** |
> | Alpaca WR | 34.72 | 38.14 | 37.28 | **39.60** |
>
> As shown in the table, our $M_{AP}$ metric continuously outperforms other methods **across different proportions of selected data**.
>
> For the bottom-40% selection, data was reversely chosen using various metrics from the Llama v2 dataset for SimPO training:
>
> | **Reverse 40%** | Unifrom | $M_r$ | $M_\pi$ | $M_{AP}$ |
> |---|:---:|:---:|:---:|:---:|
> | Alpaca LC | 41.61 | 40.66 | 37.07 | _34.64_ |
> | Alpaca WR | 36.26 | 33.36 | 32.95 | _29.37_ |
>
> Across all metrics—when selecting the bottom 40% based on $M_r, -M_\pi, M_{AP}$—the performance of resultant models was poorer than the uniform baseline, indicating lower data quality through reverse selection.
> Notably, **the lowest performance** observed was with our $M_{AP}$ metric, verifying its efficacy in identifying high-quality datasets **from an opposite standpoint**.
>
> These experiments affirm the effectiveness of our proposed method, and we will incorporate the findings in our revised version.
>
> ## More Iterative Baselines & References
>
> Thank you for your valuable suggestions!
>
> In the iterative preference learning experiments, we augment the default iterative preference learning pipeline by introducing an additional data selection procedure guided by the proposed metric $M_{AP}$.
> The role of our proposed metric in this process is to **select high-quality preference data** for subsequent training.
> Therefore, it serves as an **orthogonal strategy** alongside existing iterative preference optimization techniques [1,2,3].
> While integrating our data selection process with these optimization methods poses interesting prospects, we're afraid the current timeframe does not allow us to explore all potential directions.
>
> We sincerely appreciate you for highlighting these relevant works.
> We will include them in our references and discuss potential integrations of our $M_{AP}$ metric with various iterative preference learning methods in the future work section of our revised version.

---

### Decision · Program_Chairs · 2025-05-01

**Decision:**

Accept (poster)

**Comment:**

This paper received 6 reviews since some originally-assigned reviewers submitted the review after the deadline. After carefully checking the paper, the reviews, the rebuttal, and the author-reviewer discussions, I think the strong points outweigh the weak points, which can be fixed in the camera-ready version. Thus, I recommend accepting this paper.